# The impact of internet use on the subjective well-being of older adults: The mediating role of mental health

Jiangwei Hu, Chunyun Tan*

School of Chinese Literature and Media, Hubei University of Arts and Science, Xiangyang, Hubei, China

* pioneer928@163.com

## Abstract

### Background

As internet use rises among older adults, the internet has become a vital tool for maintaining social ties and enhancing life satisfaction. Prior research suggests that online engagement may be linked to subjective well-being (SWB) by offering emotional support and opportunities for participation. However, the psychological pathways underlying this association—such as psychological anxiety, social loneliness, and goal deficiency (reduced sense of purpose)—remain underexplored. To address this gap, this study examines how these psychological factors are associated with internet use and older adults' SWB in a cross-sectional context.

### Methods

Drawing on cross-sectional data from the 2021 Chinese General Social Survey (CGSS), this study analyzed a sample of 825 Chinese adults aged 60 years and above. A structural equation modeling (SEM) approach was used to examine the associations among internet use, three psychological variables (psychological anxiety, social loneliness, and goal deficiency), and SWB. Control variables included age, gender, education level, and place of residence.

### Results

The direct association between internet use and SWB was non-significant. However, internet use was associated with lower psychological anxiety, lower social loneliness, and lower goal deficiency, and the overall indirect association with SWB was positive. Among the mediators, psychological anxiety accounted for the largest share of the indirect association, goal deficiency contributed modestly, and the loneliness pathway was not statistically significant. The total association between internet use and SWB remained positive when indirect paths were considered.

**Data availability statement:** The data used in this study are derived from the 2021 wave of the Chinese General Social Survey (CGSS), administered by the National Survey Research Center at Renmin University of China. The dataset is not publicly downloadable, but it is accessible to academic researchers upon request. Interested users may obtain the data by registering and applying through the Chinese National Survey Data Archive (CNSDA) at the following link: http://www.cnsda.org/index. php?r=projects/view&id=65635422. Access is granted after completing a data use application and stating the research purpose. The authors had no special access privileges and obtained the data through the same procedure available to any qualified researcher.

**Funding:** The author(s) received no specific funding for this work.

**Competing interests:** The authors have declared that no competing interests exist.

## Conclusion

The evidence indicates that, among older adults, digital engagement is associated with SWB chiefly via mental-health pathways—particularly through its associations with lower anxiety and reduced goal deficiency—rather than through a direct association. These findings suggest that policy and practice may complement access and digital-literacy initiatives with supports that reduce anxiety and strengthen purpose and competence (e.g., step-by-step onboarding, simplified interfaces, peer mentoring) as well as goal-oriented uses such as health self-management and community participation. Since the loneliness-mediated route was not supported, programs should emphasize emotionally meaningful online connections and relationship quality rather than merely increasing contact volume.

## Introduction

The global aging trend has accelerated in recent years. As of 2020, there were 1 billion people aged 60 and above globally, projected to reach 2 billion by 2050 [1]. In China, the aging population has similarly expanded, with 264 million older adults in 2020 (18.9% of the total population), expected to surpass 300 million by 2025 [2]. Concurrently, the rapid growth of internet access has provided older adults with new channels for social participation and engagement. In 2021, 43.2% of older adults in China had access to the internet, reflecting their increasing involvement in digital society [3]. Internet use has become an increasingly important facet of life among older adults, offering avenues for information access, emotional expression, and social engagement [4]. A growing body of literature suggests that internet use can enhance subjective well-being by reducing loneliness and facilitating social connectivity [5,6]. Theoretically, Maslow's hierarchy of needs proposes that digital interaction can help older adults fulfill higher-order needs—such as belongingness and self-esteem—through online connection and expression [7,8], while activity theory asserts that participation in meaningful activities can alleviate negative emotions like loneliness and improve life satisfaction [9]. These frameworks provide a foundation for examining how digital experiences shape well-being.

However, empirical evidence on the relationship between internet use and well-being remains mixed. Çikrıkci [10], through a meta-analysis, showed that the effects of internet use on well-being vary substantially depending on users' psychological states, usage patterns, and demographic characteristics. Similarly, Heo found that while internet use can reduce loneliness among older adults, it may not always enhance emotional well-being, especially in the absence of quality offline interactions [11]. These findings highlight the dual nature of internet use—both beneficial and potentially detrimental—depending on contextual factors. Given the rapid digitalization of society and the increasing adoption of internet technology by older adults, it is important to understand how these demographic experiences digital engagement. Older adults exhibit distinctive patterns of internet use, primarily oriented toward maintaining family ties, obtaining health-related information, and alleviating loneliness

[12]. They also face elevated mental health risks due to life-course transitions that shrink social networks and promote social exclusion [13]. Moreover, their subjective well-being is particularly sensitive to variations in emotional state and social support, with perceived support strongly predicting life satisfaction across ages [14]. These characteristics make them a critical group for examining how internet use influences well-being via psychological mechanisms, such as loneliness reduction and emotional resilience.

Mental health, as a fundamental component of quality of life, becomes particularly salient in later life [15]. Older adults are especially sensitive to psychological vulnerabilities, as they often experience transitions such as retirement, bereavement, and social role reduction, which may affect their emotional and cognitive well-being [16]. From the perspective of psychological pathway mechanisms, internet use can impact subjective well-being through emotional regulation and social cognitive processes. Socioemotional Selectivity Theory (SST) posits that older adults prioritize emotionally meaningful goals and social interactions, which digital platforms may facilitate [17]. Similarly, Life Course Theory underscores how age-related transitions reshape individuals' social needs and coping resources, reinforcing the role of mental health in adapting to technological environments [18]. Existing studies suggest that moderate internet use may offer emotional support and social opportunities for older adults, thereby alleviating negative emotions such as anxiety and loneliness and enhancing their psychological well-being. However, excessive or maladaptive use may result in cognitive overload and increased psychological stress [19]. Furthermore, Social Compensation Theory posits that individuals with limited offline social resources may turn to online interactions as a compensatory strategy to meet their emotional and social needs [20]. This compensatory use may vary based on mental health status, reinforcing its mediating role.

Despite growing scholarly interest in how internet use affects the subjective well-being of older adults, existing studies have primarily concentrated on its direct effects, with insufficient attention to the underlying psychological mechanisms. In particular, the mediating role of mental health remains underexplored. Although previous research has shown associations between internet use and emotional outcomes [11,21], few studies have systematically investigated how specific psychological constructs—such as anxiety, loneliness, or goal-related cognition—function as mediators in this relationship [22]. Addressing this gap is essential for advancing theoretical understanding and informing targeted mental health interventions in digitally connected aging populations.

Accordingly, this study examines the association between internet-use frequency and subjective well-being (SWB) among older adults, with a specific focus on the mediating role of mental health. Using nationally representative data from the China General Social Survey (CGSS), we test a multiple-mediator model in which three psychological dimensions—psychological anxiety, social loneliness, and goal deficiency (purpose in life)—transmit the association between internet-use frequency and SWB. The proposed framework (Fig 1) motivates our hypotheses and guides the subsequent empirical analysis.

## Literature review

### Internet use and subjective well-being

The widespread adoption of the internet in the Web 2.0 era has opened up extensive opportunities for social engagement, information exchange, and emotional interaction among users [23]. For older adults, the internet facilitates sustained connections with family and friends, providing emotional support and a sense of belonging [24]. Prior studies have generally affirmed that such interactions may enhance life satisfaction and reduce loneliness, thereby promoting subjective well-being. However, the effects of internet use are not unilaterally positive. While active engagement online can foster emotional pleasure and social fulfillment, passive or aimless browsing may instead induce negative emotions or upward social comparison [25]. Flow theory further explains this duality by suggesting that immersive and meaningful digital experiences can generate satisfaction and self-accomplishment among older users [26].

Building on these theoretical perspectives, recent empirical evidence has consistently supported the positive association between internet use and subjective well-being among older adults. Using nationally representative data from China,

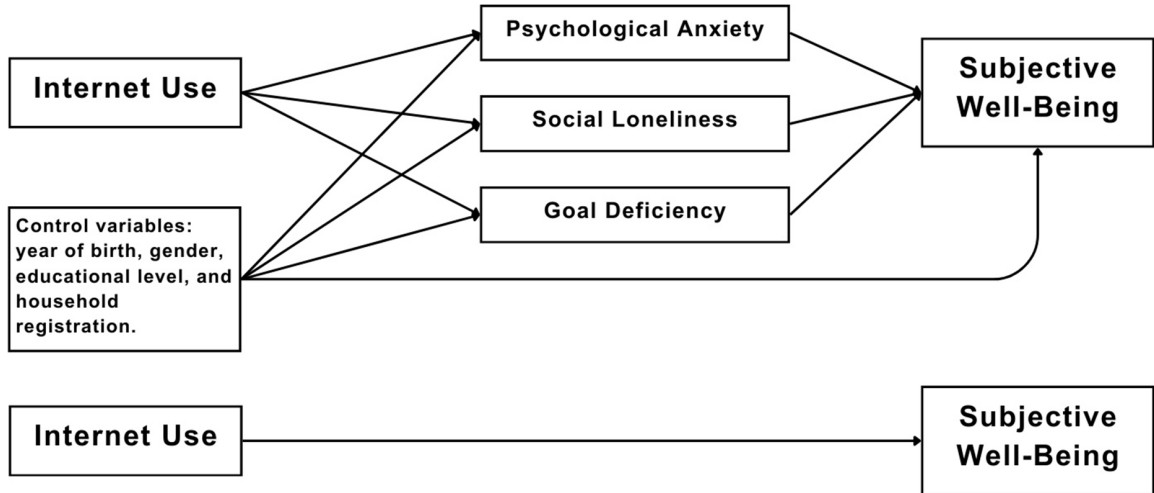

**Fig 1. The Hypothetical Model.**

one study [27] demonstrated that internet use frequency, the size of online social networks, and digital proficiency each had significant positive effects on subjective well-being among middle-aged and older adults. Another study [28] on older adults' internet activities reported that communicative uses of the internet exert stronger positive effects on eudaimonic well-being than passive or purely informational uses. Similarly, a comparative study conducted in Finland and Sweden [29] found that socially oriented and active internet use was positively related to subjective well-being and psychological health among older populations. Collectively, these studies provide a solid basis for the following hypothesis:

H1: Internet use is positively associated with subjective well-being among older adults.

Despite these insights, existing research has primarily emphasized the outcomes of internet use on well-being, while neglecting the internal psychological processes through which such effects occur. Specifically, the role of psychological states—such as anxiety, loneliness, and goal deficiency—has received limited attention as potential mediators. This study therefore seeks to extend prior findings by investigating whether internet use influences subjective well-being through psychological mechanisms, thereby providing a more nuanced understanding of how digital participation shapes older adults' emotional experiences.

### Internet use and mental health

The relationship between internet use and mental health among older adults is multifaceted. In this study, mental health is conceptualized as a multidimensional construct encompassing three aspects: psychological anxiety, social loneliness, and goal deficiency. These dimensions reflect emotional distress, weakened social connectedness, and diminished sense of purpose in later life. On the one hand, engaging in online interactions has been found to reduce depressive symptoms, strengthen psychological resilience, and mitigate loneliness [30]. These benefits are largely attributed to the internet's ability to provide instant access to emotional support, companionship, and relevant health information, enabling older individuals to cope more effectively with stressors. The interactive and reciprocal nature of digital communication has also been shown to promote a sense of belonging, which serves as a buffer against psychological distress and enhances overall well-being [12].

However, these benefits are not universal. Excessive or unregulated internet use may lead to information overload, social comparison, and reduced real-life interaction, potentially heightening psychological anxiety and emotional exhaustion [19]. This phenomenon is often framed as digital stress—the strain produced by persistent digital demands such as

continuous notifications, information overload, and exposure to misinformation—factors qualitatively linked to anxiety in older adults [31] and assessed with instruments like the Digital Stressors Scale [32].

At the same time, age-related changes can erode social roles and network structure; as Weiss's relational loneliness framework highlights, older adults are therefore especially susceptible to isolation, underscoring the salience of the loneliness pathway in later life [33]. Grounded in Self-Determination Theory, psychological well-being depends on a sustained sense of purpose and competence; accordingly, overall internet-use frequency can shape affective and relational states that support—or undermine—these needs [34,35]. Consistent with Socioemotional Selectivity Theory, older adults prioritize emotionally meaningful ties, so regular online contact may help mitigate loneliness and related affective outcomes [17]. From a life-course perspective, transitions such as retirement or bereavement alter social integration and coping resources. In this context, internet-use frequency provides a parsimonious indicator of exposure to supportive—or potentially depleting—online interactions [18].

Empirical evidence aligns with this framework: higher internet-use frequency is associated with lower loneliness and greater well-being in older adults [11,12] and can buffer the adverse effects of social isolation on depression and cognition [16]. Accordingly, we treat internet-use frequency as the exposure variable and test whether three mental-health dimensions—psychological anxiety, social loneliness, and goal deficiency—mediate its association with subjective well-being in later life. Based on this framework and prior evidence, we posit the following direct associations between internet use and the three mental-health dimensions:

H2: Internet use is negatively associated with psychological anxiety.
H3: Internet use is negatively associated with social loneliness.
H4: Internet use is negatively associated with goal deficiency.

### Mental Health and Subjective Well-Being

Mental health is a well-established determinant of subjective well-being (SWB), particularly among older adults, who are more vulnerable to emotional instability and life stressors in later life. SWB typically reflects individuals' cognitive and affective evaluations of their life quality [36]. Empirical studies have shown that older adults with better psychological health tend to report higher levels of SWB [37] not merely due to the absence of mental disorders, but also through their enhanced capacity to positively interpret life events [38]. However, the construct of mental health is multifaceted, encompassing dimensions such as psychological anxiety, social loneliness, and goal deficiency. These psychological states can function as mediators that link external experiences—such as digital engagement—to internal emotional evaluations like well-being. For instance, older adults experiencing lower anxiety and loneliness are more likely to benefit from online interactions, while those who lack a sense of life purpose may not derive meaningful satisfaction from internet use [39].

Therefore, this study conceptualizes psychological anxiety, social loneliness, and goal deficiency as key mediating variables that bridge the relationship between internet use and subjective well-being. By unpacking the specific mechanisms through which these mental health dimensions operate, this research aims to extend existing knowledge on how digital behaviors interact with psychological profiles to shape emotional outcomes in late adulthood. Building on this rationale and the preceding evidence, we formally propose the following multiple-mediator hypothesis:

H5: Internet use is associated with subjective well-being indirectly through mental health.
Fig 1 illustrates the hypothesized mediation model.

## Methods

### Data

The data for this study were derived from the 2021 Chinese General Social Survey (CGSS 2021), the first nationwide, comprehensive, and continuous academic survey project in mainland China. It is hosted by Renmin University of China and implemented by the China Survey Data Center. Since its inception in 2003, this project has conducted

annual cross-sectional surveys of over 10,000 households across the country, systematically collecting data at multiple levels, including individuals, families, and communities. It has provided critical support for social science research and policymaking.

The CGSS 2021 dataset includes key variables relevant to this study, such as the frequency of internet use among older adults, subjective well-being (SWB), and mental health indicators (e.g., anxiety, loneliness, and goal deficiency). This study focuses on individuals aged 60 and above, yielding a total of 825 valid samples, providing a robust scientific foundation for exploring the relationship between internet use and subjective well-being among older adults.

## Measure

### Variable.

(1) **Dependent Variable: Subjective Well-Being (SWB)**

The dependent variable, Subjective Well-Being (SWB), is defined as respondents' subjective evaluation of their current level of life happiness. Drawing on the research approaches of Liu et al. [40] and Ding et al. [41], this study uses the question from the 2021 CGSS questionnaire: "Overall, do you feel happy with your life?" as the basis for measurement. Respondents' answers, ranging from "very unhappy" to "very happy," are assigned scores from 1 to 5, with higher scores indicating stronger subjective well-being.

(2) **Independent Variable: Internet Use**

The independent variable is internet use. Drawing on the research framework proposed by Xu and Zhang [42], this study uses the question from the 2021 CGSS questionnaire: "How often did you use the internet over the past year?" as the basis for measurement. Respondents' answers, ranging from "never" to "always," are assigned scores from 1 to 5, with higher scores indicating greater internet use frequency.

(3) **Mediating Variable: Mental Health**

In this study, mental health is conceptualized as three dimensions—psychological anxiety, social loneliness, and goal deficiency. Following Deng [43], we used three items from the 2021 China General Social Survey (CGSS) to measure these constructs: (a) Psychological anxiety: "In the past four weeks, how often did you feel anxious?" (1 = not at all; 5 = extremely strong). (b) Social loneliness: "In the past year, how often did you feel lonely?" (1 = never; 5 = always). (c) Goal deficiency: "In the past year, did you feel a lack of purpose in life?" (1 = not at all; 5 = extremely serious). These variables were specified as mediators in the structural equation model (SEM) to examine whether mental health serves as a pathway through which internet use affects subjective well-being among older adults.

(4) **Control variables**

Existing research has shown that the measurement of subjective well-being is influenced by individual background characteristics [44–47]. To control for other factors that may affect subjective well-being, this study includes the following control variables: Year of Birth: Adjusts for potential age-related differences. Gender: Coded as female = 0, male = 1. Education Level: Categorized based on the highest level of education attained, with scores ranging from 1 to 5 representing "primary school or below," "middle school," "high school," "college or equivalent," and "graduate or above," respectively. Place of Residence: Coded as rural = 0, urban = 1.

(5) **Statistical analysis**

This study conducted statistical analyses in the following steps: First, descriptive statistics were performed, including means and standard deviations for continuous variables, and frequencies and percentages for categorical variables,

to present sample characteristics and variable distributions. Second, Pearson correlation analysis was conducted to preliminarily assess the relationships among key variables, including internet use, mental health factors, and subjective well-being. Finally, structural equation modeling (SEM) was employed to examine both the direct and indirect effects of internet use on subjective well-being through the mediating roles of psychological anxiety, social loneliness, and goal deficiency. The significance of indirect effects was tested using the bias-corrected bootstrap method with 5,000 resamples and 95% confidence intervals. Results were considered statistically significant if the confidence interval did not include zero.

## Results

### Descriptive statistics

Table 1 presents the descriptive statistics and demographic characteristics of the sample. The average year of birth was 1951 (*SD = 6.586, range: 1929–1961*), with 51.4% of participants being female and 48.6% male. In terms of education, 47.3% had primary education or below, 29.9% had completed middle school, 17.2% high school, 5.4% held a college degree or above, and only 0.2% had received graduate-level education. The mean score for subjective well-being was 3.6 (*SD = 0.987*), with 36.6% reporting being "somewhat happy," 18.7% "very happy," 34.5% neutral, 6.3% "somewhat unhappy," and 3.9% "very unhappy." Internet use had a mean score of 2.15 (*SD = 1.552*); specifically, 58.4% reported never using the internet, 9.2% rarely, 5.3% occasionally, 12.8% often, and 14.2% frequently. The average score for psychological anxiety was 2.56 (*SD = 1.266*), with 23.9% feeling "extremely" anxious, 32.1% "very much," 15.2% "moderately," 21.3% "slightly," and 7.5% "not at all." The mean score for social loneliness was 2.22 (*SD = 1.097*), and for goal deficiency 2.44 (*SD = 1.169*), with response distributions similarly skewed toward higher intensity categories. Regarding place of residence, 94.3% of the participants lived in rural areas, while only 5.7% were from urban areas.

### Correlation analysis

As shown in Table 2, most correlations among the key study variables were statistically significant and aligned with expectations. Subjective well-being correlated positively with internet use (*r = .10, p < .01*) and education level (*r = .13, p < .01*), and most strongly and negatively with psychological anxiety (*r = −.27, p < .01*), followed by social loneliness and goal deficiency. Internet use was also consistently linked to lower levels of psychological anxiety, social loneliness, and goal deficiency, and was more prevalent among younger and better-educated individuals. By contrast, non-significant correlations, such as those with gender and place of residence, were minimal and are reported in Table 2. Overall, these results highlight robust patterns of association but, given the cross-sectional design, do not imply causal relationships.

### The relationship between internet use and subjective well-being among older adults

Before the mediation analysis (Table 4), a series of stepwise OLS regression models were conducted to examine how internet use and psychological factors relate to subjective well-being. These models (Table 3) offer foundational insight and complement the structural equation model by highlighting changes in explanatory power as variables are introduced sequentially. The specific analysis of each model is as follows:

### Model 1: Effects of control variables

The regression including only control variables shows that year of birth (*B = −0.014, SE = 0.005, p < 0.01*) is negatively associated with subjective well-being, indicating that older individuals tend to report slightly higher levels of well-being. Education level (*B = 0.150, SE = 0.038, p < 0.001*) has a positive and significant effect. Gender (*B = 0.035, p = 0.610*) and place of residence (*B = 0.047, p = 0.754*) are not significantly associated with well-being.

**Table 1. Descriptive statistics of key variables.**

| Variables | Mean/percentage | Sd | Min | Max | details |
|---|---|---|---|---|---|
| **Subjective Well-Being** | 3.6 | 0.987 | 1 | 5 | Multi-categorical variables |
| Very Unhappy | 3.9% | | | | |
| Somewhat Unhappy | 6.3% | | | | |
| Neutral (Neither Happy nor Unhappy) | 34.5% | | | | |
| Somewhat Happy | 36.6% | | | | |
| Very Happy | 18.7% | | | | |
| **Internet Use** | 2.15 | 1.552 | 1 | 5 | Multi-categorical variables |
| Never | 58.4% | | | | |
| Rarely | 9.2% | | | | |
| Occasionally | 5.3% | | | | |
| Often | 12.8% | | | | |
| Frequently | 14.2% | | | | |
| **Psychological Anxiety** | 2.56 | 1.266 | 1 | 5 | Multi-categorical variables |
| Extremely | 23.9% | | | | |
| Very much | 32.1% | | | | |
| Moderately | 15.2% | | | | |
| Slightly | 21.3% | | | | |
| Not at all | 7.5% | | | | |
| **Social Loneliness** | 2.22 | 1.097 | 1 | 5 | Multi-categorical variables |
| Extremely | 27.0% | | | | |
| Very much | 44.5% | | | | |
| Moderately | 11.5% | | | | |
| Slightly | 13.2% | | | | |
| Not at all | 3.8% | | | | |
| **Goal Deficiency** | 2.44 | 1.169 | 1 | 5 | Multi-categorical variables |
| Extremely | 23.3% | | | | |
| Very much | 38.1% | | | | |
| Moderately | 14.1% | | | | |
| Slightly | 20.6% | | | | |
| Not at all | 4.0% | | | | |
| **Year of Birth** | 1951 | 6.586 | 1929 | 1961 | Continuous variables |
| **Gender** | 0.49 | 0.500 | 0 | 1 | Binary variables |
| Female | 51.4% | | | | |
| Male | 48.6% | | | | |
| **Education Level** | 1.81 | 0.921 | 1 | 5 | Multi-categorical variables |
| Primary Education or Below | 47.3% | | | | |
| Middle School Education | 29.9% | | | | |
| High School Education | 17.2% | | | | |
| College or Above | 5.4% | | | | |
| Graduate or Above | 0.2% | | | | |
| **Place of Residence** | 0.06 | 0.232 | 0 | 1 | Binary variables |
| Rural | 94.3% | | | | |
| Urban | 5.7% | | | | |

**Table 2. Pearson's correlations among relevant study variables.**

| Variables | 1 | 2 | 3 | 4 | 5 | 6 | 7 | 8 | 9 |
|---|---|---|---|---|---|---|---|---|---|
| 1. Subjective Well-Being | 1 | | | | | | | | |
| 2. Internet Use | 0.104** | 1 | | | | | | | |
| 3. Psychological Anxiety | −0.270** | −0.126** | 1 | | | | | | |
| 4. Social Loneliness | −0.193** | −0.073* | 0.266** | 1 | | | | | |
| 5. Goal Deficiency | −0.200** | −0.100** | 0.259** | 0.182** | 1 | | | | |
| 6. Year of Birth | −0.071* | 0.200** | 0.054 | −0.044 | −0.002 | 1 | | | |
| 7. Gender | 0.046 | −0.011 | −0.166** | 0.071* | −0.057 | −0.042 | 1 | | |
| 8. Education Level | 0.132** | 0.403** | −0.149** | −0.080* | −0.040 | 0.134** | 0.166** | 1 | |
| 9. Place of Residence | 0.020 | 0.111** | 0.014 | −0.016 | −0.016 | 0.061 | 0.002 | 0.136** | 1 |

$*p < 0.05; **p < 0.01; ***p < 0.001$

**Table 3. Stepwise OLS regression analysis of the impact of internet use and mental health on subjective well-being.**

| | Model 1 | Model 2 | Model 3 | Model 4 | Model 5 |
|---|---|---|---|---|---|
| Year of Birth | −0.014** | −0.016** | −0.013* | −0.014** | −0.014** |
| | (0.005) | (0.005) | (0.005) | (0.005) | (0.005) |
| Gender | 0.035 | 0.046 | −0.026 | 0.004 | −0.005 |
| | (0.069) | (0.069) | (0.068) | (0.068) | (0.067) |
| Education Level | 0.150*** | 0.118** | 0.092* | 0.085* | 0.091* |
| | (0.038) | (0.041) | (0.040) | (0.040) | (0.040) |
| Place of Residence | 0.047 | 0.034 | 0.071 | 0.061 | 0.055 |
| | (0.149) | (0.149) | (0.145) | (0.144) | (0.143) |
| Internet Use | | 0.049* | 0.032 | 0.032 | 0.025 |
| | | (0.024) | (0.024) | (0.024) | (0.023) |
| Psychological Anxiety | | | −0.193*** | −0.164*** | −0.143*** |
| | | | (0.027) | (0.028) | (0.028) |
| Social Loneliness | | | | −0.118*** | −0.103*** |
| | | | | (0.031) | (0.031) |
| Goal Deficiency | | | | | −0.107*** |
| | | | | | (0.029) |
| Constant | 30.620** | 33.851*** | 28.753** | 30.553** | 30.772** |
| | (10.201) | (10.309) | (10.026) | (9.957) | (9.881) |
| N | 819 | 819 | 819 | 819 | 819 |
| R² | 0.027 | 0.031 | 0.089 | 0.105 | 0.120 |

$*p < 0.05; **p < 0.01; ***p < 0.001.$

## Model 2: Inclusion of internet use

Adding internet use reveals a significant positive relationship with subjective well-being ($B = 0.049$, $SE = 0.024$, $p < 0.05$). The coefficients for year of birth ($B = −0.016$, $p < 0.01$) and education level ($B = 0.118$, $p < 0.01$) remain significant. Gender and residence continue to show no meaningful effects.

**Table 4. Path Coefficients for Direct, Indirect, and Total Effects of Internet Use and Psychological Factors on Subjective Well-being.**

| Path | Standard Estimate | Standard Error | z-value | P-value | 95% CI |
|---|---|---|---|---|---|
| **Direct effect** | | | | | |
| Internet Use → Psychological Anxiety | −0.126 | 0.034 | −3.660 | <0.001 | [−0.158, −0.048] |
| Internet Use → Social Loneliness | −0.073 | 0.035 | −2.108 | 0.035 | [−0.100, −0.004] |
| Internet Use → Goal Deficiency | −0.100 | 0.035 | −2.879 | 0.004 | [−0.126, −0.024] |
| Internet Use → Subjective Well-being | 0.059 | 0.034 | 1.738 | 0.082 | [−0.005, 0.079] |
| Psychological Anxiety → Subjective Well-being | −0.204 | 0.034 | −6.059 | <0.001 | [−0.207, −0.106] |
| Social Loneliness → Subjective Well-being | −0.115 | 0.033 | −3.440 | 0.001 | [−0.160, −0.044] |
| Goal Deficiency → Subjective Well-being | −0.124 | 0.034 | −3.683 | <0.001 | [−0.158, −0.048] |
| **Indirect effect** | | | | | |
| Internet Use → PA → Subjective Well-being | 0.016 | 0.005 | 3.133 | 0.002 | [0.006, 0.026] |
| Internet Use → SL → Subjective Well-being | 0.005 | 0.003 | 1.798 | 0.072 | [0.000, 0.011] |
| Internet Use → GD → Subjective Well-being | 0.008 | 0.004 | 2.268 | 0.023 | [0.001, 0.014] |
| Total indirect effect | 0.029 | 0.007 | 4.261 | <0.001 | [0.016, 0.043] |
| **Total effect** | | | | | |
| Internet Use → Subjective Well-being (Total) | 0.066 | 0.022 | 3.047 | 0.002 | [0.024, 0.109] |

PA = Psychological Anxiety; SL = Social Loneliness; GD = Goal Deficiency

Note: The total effect (0.066) differs slightly from the sum of the direct (0.059) and indirect (0.029) effects because standardized SEM coefficients are estimated separately and may not add up exactly. This does not affect result validity.

## Model 3: Inclusion of psychological anxiety

Psychological anxiety enters as a strong negative predictor ($B=−0.193$, $SE=0.027$, $p<0.001$). With its inclusion, the effect of internet use becomes statistically non-significant ($B=0.032$, $p=0.183$), suggesting a possible mediating role of anxiety. Year of birth ($B=−0.013$, $p<0.05$) and education ($B=0.092$, $p<0.05$) remain significant; gender and residence remain non-significant.

## Model 4: Inclusion of social loneliness

Social loneliness also shows a significant negative relationship with well-being ($B=−0.118$, $p<0.001$). Anxiety ($B=−0.164$, $p<0.001$) remains significant, while internet use stays non-significant ($B=0.032$, $p=0.179$). Year of birth and education remain significant.

## Model 5: Inclusion of goal deficiency

Goal deficiency is significantly negatively associated with well-being ($B=−0.107$, $SE=0.029$, $p<0.001$). Psychological anxiety ($B=−0.143$, $p<0.001$) and social loneliness ($B=−0.103$, $p<0.01$) continue to show meaningful effects. Internet use again does not exert a direct effect ($B=0.025$, $p=0.284$), supporting the notion that its impact may primarily operate through indirect psychological pathways. Year of birth and education maintain their positive associations; gender and residence remain non-significant.

## Mediating effect analysis

To further examine the mechanism through which internet use affects subjective well-being, a structural equation modeling (SEM) approach was applied, and the standardized path coefficients are presented in Table 4 and Fig 2. The results reveal that internet use has significant indirect effects on subjective well-being through three psychological variables—psychological anxiety, social loneliness, and goal deficiency. These findings suggest that the relationship between internet use and well-being is largely mediated by individuals' mental health status.

As shown in Table 4, the direct effect of internet use on subjective well-being is not statistically significant ($β=0.059$, $p=0.082$), so H1 is not supported.

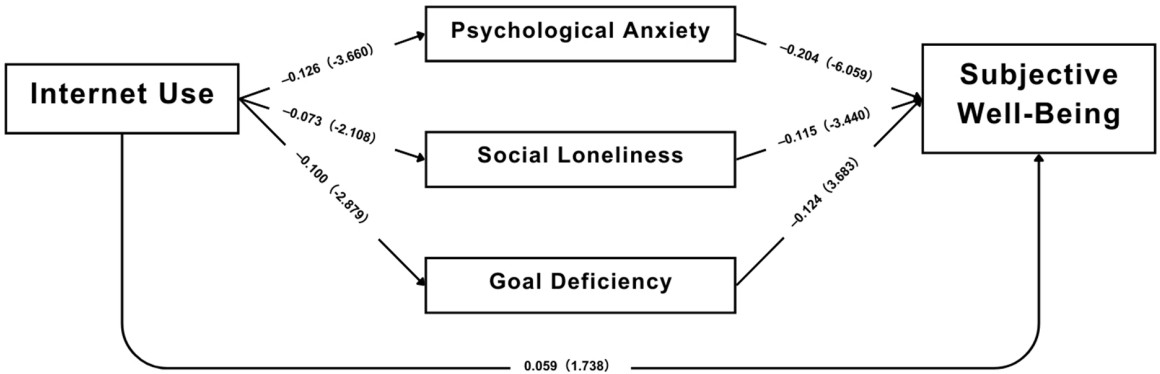

**Fig 2. Standardized path coefficients and corresponding t statistics.**

Turning to the proposed mediating pathways, internet use is significantly associated with lower psychological anxiety (*β = −0.126, p < 0.001*), lower social loneliness (*β = −0.073, p = 0.035*), and lower goal deficiency (*β = −0.100, p = 0.004*), supporting H2–H4 (direct paths). In turn, higher levels of these three psychological variables predict lower subjective well-being (*β range = −0.115 to −0.204*), However, the product-of-coefficients tests show that the indirect effect via social loneliness is not significant, whereas the indirect effects via psychological anxiety and goal deficiency are significant. Thus, the overall indirect effect is supported, but the loneliness pathway itself is not.

Building on these findings, the mediation analysis (H5) indicates that the total indirect effect is significant (*β = 0.029, p < 0.001*), suggesting that internet use is associated with well-being primarily through mental-health pathways. At the level of specific mediators, the indirect effect via psychological anxiety is significant (*β = 0.016, p = 0.002*) and via goal deficiency is significant (*β = 0.008, p = 0.023*); the pathway via social loneliness is marginal (*β = 0.005, p = 0.072*) and thus does not reach the conventional 0.05 level. Taken together, these findings provide overall support for H5, with mixed evidence across individual mediators.

Finally, the overall model fit was assessed. Goodness-of-fit indices are reported in Table 5 and were evaluated against commonly recommended thresholds [48]—$\chi^2$/df < 3, RMSEA ≤ 0.06, CFI ≥ 0.95, SRMR ≤ 0.08 (with GFI/AGFI/NFI/NNFI/IFI ≥ 0.90 used as conventional heuristics).

## Discussion

This study examined how internet use relates to subjective well-being (SWB) among older adults, with particular attention to the mediating roles of psychological anxiety, social loneliness, and goal deficiency. Our findings contribute to the literature by clarifying that the link between internet use and SWB is predominantly indirect, operating through specific psychological pathways rather than via a direct association. In this sense, our study extends prior research by incorporating goal deficiency alongside anxiety and loneliness, thus providing a more comprehensive account of the mental-health mechanisms underlying digital engagement in later life.

Contrary to H1, the direct association between internet use and SWB was not statistically significant. At the same time, the indirect associations were significant, indicating that the influence of internet use on SWB operates primarily through psychological mechanisms rather than a direct path. This pattern aligns with a mediated (indirect-only) account of the

**Table 5. Fit Indices of the structural equation model.**

| Index | $\chi^2$/df | RMSEA | CFI | SRMR | GFI | AGFI | NFI | NNFI | IFI |
|---|---|---|---|---|---|---|---|---|---|
| Acceptable values | < 3 | < 0.06 | > 0.95 | < 0.08 | > 0.90 | > 0.90 | > 0.90 | > 0.90 | > 0.90 |
| Observed values | 2.5 | 0.06 | 0.95 | 0.07 | 0.95 | 0.92 | 0.93 | 0.92 | 0.94 |

relationship. Importantly, it also resonates with prior research, which emphasizes that the benefits of digital participation for well-being are realized primarily through psychological or social mechanisms, a conclusion increasingly supported in studies of older adults [49]. These findings suggest that internet use alone may not guarantee better well-being outcomes without improvements in mental-health conditions.

Regarding individual mediators, the results support H2 and H4: internet use is negatively associated with psychological anxiety and with goal deficiency, and each of these variables in turn predicts lower SWB—consistent with their roles as mechanisms. Together they account for the bulk of the overall mediation effect and support the view that digital engagement can be linked to better well-being when it is accompanied by reduced anxiety and a stronger sense of purpose or competence [20]. For H3, two conclusions should be distinguished. The direct link between internet use and social loneliness is negative and significant, indicating that more frequent online contact can be associated with reduced perceived loneliness [50]. However, the indirect effect of internet use on SWB through loneliness is not significant at conventional levels. A plausible interpretation is that increased contact does not automatically translate into improvements in the quality or emotional depth of relationships—particularly for older adults facing barriers such as limited digital literacy. Thus, loneliness appears less central as a mediating route in this dataset compared with anxiety and purpose-related processes.

Taken together, the pattern supports H5: mental-health variables collectively account for the association between internet use and SWB. The evidence suggests an internal pathway in which digital engagement is associated with well-being, largely through its links with lower anxiety and stronger goal-related motivation, rather than via a direct effect on life evaluations. This finding complements prior work by Zhang, Shi, and Feng [51], who demonstrated that the effects of internet use on well-being are stratified by subjective social class, with psychological resources—rather than direct pathways—playing a central mediating role. Our study therefore adds weight to the argument that digital participation enhances well-being primarily when it strengthens psychological resources and goal orientation.

From a practical perspective, this mediated pattern has implications for intervention. Programs should target the mediators themselves—reducing anxiety, strengthening purpose, and supporting meaningful ties—while making digital engagement simpler and more goal-directed for older adults. Concretely, this entails making use manageable (step-by-step onboarding, simplified interfaces, peer mentors/helplines), meaningful (goal-oriented activities such as health self-management, intergenerational messaging, community micro-volunteering, hobby groups), and social and safe (small, moderated groups; regular video circles with privacy/anti-fraud cues). Interventions should also be tailored to different levels of digital literacy and mental-health profiles, with scaffolded modules for low-literacy users and purpose-building tasks where goal orientation is weak.

Several limitations should be noted. Measurement relied on single-item self-reports; future work should employ validated multi-item scales or behavioral log data. Design was cross-sectional; longitudinal or experimental approaches are needed to establish temporal order. Sample composition in our older-adult subsample is rural-heavy, reflecting CGSS's hukou-based residence classification and cohort-specific urbanization; this may limit generalization to urban older adults. Applying population weights where appropriate can better align estimates with national distributions. Finally, unobserved factors (e.g., personality, major life events, offline networks) were not modeled and warrant attention in future research, as do explicit measures of usage patterns to complement the frequency-based exposure used here.

## Conclusion

This study improves understanding of how internet use relates to older adults' subjective well-being by examining psychological anxiety, social loneliness, and goal deficiency as potential mediating pathways. The findings indicate that the mental-health benefits associated with internet use are not uniform and are realized primarily via reduced anxiety and a stronger sense of purpose, whereas the loneliness-mediated route was not supported. This pattern is consistent with a non-significant direct association between internet use and well-being, suggesting that links operate mainly through internal psychological processes rather than a direct effect on life evaluations.

From a policy standpoint, promoting digital inclusion among older adults should go beyond increasing usage frequency. It is essential to foster purposeful and interactive engagement through digital literacy programs, user-centered design, and emotionally supportive environments. Integrating mental health services—such as virtual counseling or peer support networks—may enhance well-being outcomes by reducing anxiety and reinforcing purpose/competence.

Future research should aim to improve generalizability by expanding samples to include diverse demographic groups, especially in urban contexts. Moreover, incorporating multidimensional measures of digital behavior and psychological well-being will allow for a more nuanced understanding of how technology shapes aging experiences. Ultimately, a holistic approach combining digital empowerment, emotional support, and personalized design holds the greatest potential to improve older adults' quality of life in the digital era.

## Author contributions

**Conceptualization:** chun yun Tan.

**Data curation:** chun yun Tan.

**Formal analysis:** chun yun Tan.

**Funding acquisition:** Jiangwei Hu.

**Investigation:** chun yun Tan.

**Methodology:** chun yun Tan.

**Project administration:** Jiangwei Hu.

**Resources:** Jiangwei Hu.

**Software:** chun yun Tan.

**Supervision:** Jiangwei Hu.

**Validation:** Jiangwei Hu.

**Visualization:** chun yun Tan.

**Writing – original draft:** chun yun Tan.

**Writing – review & editing:** Jiangwei Hu.

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
