## [Decision Letter · Decision Letter 0]

2 Apr 2025

Dear Dr. tan,

Thank you for submitting your manuscript to PLOS ONE. After careful consideration, we feel that it has merit but does not fully meet PLOS ONE’s publication criteria as it currently stands. Therefore, we invite you to submit a revised version of the manuscript that addresses the points raised during the review process.

**ACADEMIC EDITOR:** We have now received feedback from two experts on your manuscript. Both reviewers acknowledge the strengths of your work but also provided constructive suggestions for significant revisions. In addition to their comments, I have outlined additional points for your consideration in the section below. I hope the comments help improve the quality of this paper.

Please submit your revised manuscript by May 17 2025 11:59PM. If you will need more time than this to complete your revisions, please reply to this message or contact the journal office at plosone@plos.org . A rebuttal letter that responds to each point raised by the academic editor and reviewer(s). You should upload this letter as a separate file labeled 'Response to Reviewers'.A marked-up copy of your manuscript that highlights changes made to the original version. You should upload this as a separate file labeled 'Revised Manuscript with Track Changes'.An unmarked version of your revised paper without tracked changes. You should upload this as a separate file labeled 'Manuscript'.

We look forward to receiving your revised manuscript.

Kind regards,

Lianshan Zhang

Academic Editor

PLOS ONE

Additional Editor Comments:

We have now received feedback from two experts on your manuscript. Both reviewers recognize the merits of your work but have provided constructive suggestions for significant revisions. In addition to their comments, I also have some additional points for your consideration as you revise the manuscript.

First, since your study focuses on the mediating roles of mental health-related variables: psychological anxiety, social loneliness, and goal deficiency in the relationship between internet use and well-being, it would be beneficial to elaborate on the mediating effects of each variable rather than discussing mental health in general. As Reviewer 2 pointed out, the literature review needs to be significantly strengthened. Additionally, the rationale for conceptualizing mental health as encompassing these three specific dimensions should be well-justified with theoretical support.

Second, Tables 2–5 consistently show positive relationships among key variables. For example, in Table 3, the coefficient of 0.193 indicates that higher psychological anxiety was positively associated with subjective well-being—an unexpected finding that contradicts existing literature. If taken at face value, this result would suggest that increasing older adults’ anxiety levels could enhance their well-being, which is counterintuitive. Additionally, the internet use was positively associated with three mediators (i.e., psychological anxiety, social loneliness, and goal deficiency) as shown in Table 4. However, the discussion on page 23 interprets the findings contradictorily: “This finding aligns with the perspective of social compensation theory [36], suggesting that social media plays an especially prominent role in alleviating psychological anxiety.” Similarly, although goal deficiency was found to be positively related to subjective well-being in Table 3, the discussion on page 22 states: “Similarly, the reduction in goal deficiency also significantly mediated the relationship between social media use and subjective well-being. Social media provides crucial support for older adults in rebuilding their sense of purpose by offering opportunities to participate in community activities, express themselves, and learn new skills.” These interpretations seem inconsistent with the reported results. Please carefully review your data analysis procedures and ensure that the findings are precise and accurately reflected in the discussion.

Furthermore, regarding Table 5, the direct effect between internet use and subjective well-being is reported as 0.025, yet the total effect (direct + indirect) is the same as the total indirect effect (0.024). Please verify these statistics. Last, please report the procedures of getting the indirect effects contrast in Table 5.

Reviewers' comments:

Reviewer's Responses to Questions

**Comments to the Author**

1. Is the manuscript technically sound, and do the data support the conclusions?

Reviewer #1: Yes

Reviewer #2: Yes

2. Has the statistical analysis been performed appropriately and rigorously?

Reviewer #1: No

Reviewer #2: Yes

3. Have the authors made all data underlying the findings in their manuscript fully available?

Reviewer #1: Yes

Reviewer #2: Yes

4. Is the manuscript presented in an intelligible fashion and written in standard English?

Reviewer #1: Yes

Reviewer #2: Yes

Reviewer #1: Thank you for giving me the opportunity to review the manuscript titled "The Impact of Internet Use on the Subjective Well-Being of Older Adults: The Mediating Role of Mental Health". This manuscript addresses an important topic concerning the subjective well-being of older adults and makes a valuable contribution to the existing literature. However, I offer a few suggestions that I hope will help improve the paper.

Introduction: The introduction is well-structured and provides relevant theoretical foundations. However, more effort could be made to connect activity theory to the research topic. In fact, the theoretical foundations are not fully utilized to advance the introduction.

The literature review is well-developed.

The hypotheses are presented appropriately.

Evidence for the validity and reliability of the instruments used in the research should be provided. At least Cronbach's alpha should be reported.

Statistical analyses were performed using SPSS and the PROCESS macro. I suggest using path analysis in software such as R, LISREL, or AMOS. In this way, fit indices, which are very important in modeling, are also reported.

Assumptions of statistical analysis should be checked and reported.

Another advantage of using path analysis in the way I mentioned is that all hypotheses are tested simultaneously, and a path diagram is also reported. Your results are fragmented.

There is an error in reporting one of the sources on line 411. Please correct it.

The discussion and conclusion are well-structured.

Reviewer #2: This article, titled "The Impact of Internet Use on the Subjective Well-Being of Older Adults: The Mediating Role of Mental Health," uses data from the 2021 Chinese General Social Survey (CGSS) to explore how internet use affects the subjective well-being of older adults, with a particular focus on the mediating role of mental health. The study employs a multiple mediation analysis model and finds that internet use has a significant positive correlation with the subjective well-being of older adults, with mental health partially mediating this relationship. The findings suggest that policymakers should pay greater attention to the psychological health needs of older adults in a digital society.

Strengths of the Article

1.Relevance and Significance: The study addresses a critical and timely issue—the impact of internet use on the well-being of older adults in the context of rapid population aging and digital transformation. This topic is highly relevant to current societal challenges and policy concerns.

2.Methodological Rigor: The use of a large, nationally representative dataset (CGSS 2021) with a sample size of 825 older adults provides robust empirical support for the findings. The application of multiple mediation analysis and OLS regression models demonstrates methodological sophistication and appropriateness for the research questions.

3.Theoretical Contribution: The study contributes to the theoretical understanding of how digital participation influences the psychological well-being of older adults, offering insights into the mechanisms through which internet use affects subjective well-being.

4.Practical Implications: The findings have clear policy implications, suggesting specific interventions to enhance the digital well-being of older adults and improve their quality of life.

Major Issues

1.Literature review: the literature review is too simplistic and descriptive. It needs to be enhanced and enriched by better linking with the research hypotheses.

2.Sample Representativeness: The sample primarily consists of older adults from rural areas, which may not fully capture the characteristics of internet use among urban older adults. The generalizability of the findings to diverse populations could be questioned.

3.Measurement of Internet Use: The measurement of internet use is based on a single question about frequency, which may not capture the complexity of internet usage patterns (e.g., active vs. passive use, types of activities). A more nuanced measurement could provide deeper insights into the mechanisms at play.

4.Mediating Variables: While the study focuses on psychological anxiety, social loneliness, and goal deficiency as mediating variables, other potential mental health factors (e.g., depression, self-efficacy) could also play significant roles and should be considered in future research.

5.Discussion of Non-Significant Findings: The non-significant mediating effect of social loneliness should be discussed in greater depth, considering alternative explanations and implications for future research.

6.Policy Recommendations: The policy recommendations could be more specific, outlining concrete steps or examples of interventions that could be implemented based on the study's findings.

Minor Issues

1.Language and Clarity: Some sections of the manuscript could benefit from more concise and clear language to improve readability. For example, the abstract and introduction could be streamlined to highlight key points more effectively.

2.Data Presentation: The presentation of descriptive statistics and correlation results could be enhanced with more visual aids (e.g., tables, figures) to facilitate easier interpretation.

Overall, this study provides valuable insights into the relationship between internet use and the subjective well-being of older adults, with important implications for both theory and practice. Addressing the aforementioned issues would further strengthen the manuscript's contribution to the field.

**Do you want your identity to be public for this peer review?** For information about this choice, including consent withdrawal, please see our Privacy Policy

Reviewer #1: **Yes: ** Majid Yousefi Afrashteh

Reviewer #2: **Yes: ** Lin Zhang

---

## [Author Response · Author response to Decision Letter 1]

17 Apr 2025

Response to Academic Editor and Reviewers

Manuscript Title: The Impact of Internet Use on the Subjective Well-Being of Older Adults: The Mediating Role of Mental Health

Manuscript Number: [PONE-D-24-57555]

Dear Academic Editor and Reviewers,

On behalf of all co-authors, we would like to express our sincere gratitude for your thoughtful and constructive comments on our manuscript. We greatly appreciate the time and effort you have invested in reviewing our work. Your feedback has significantly improved the clarity, rigor, and academic contribution of the paper.

Based on your suggestions, we have made careful and substantial revisions throughout the manuscript. All changes have been marked using the Track Changes function for your review. Below, we provide a point-by-point response to each of the editor’s and reviewers’ comments. Original comments are presented in black, and our responses are in red for clarity.

We sincerely hope that the revised version meets your expectations, and we once again thank you for your valuable time and effort.

Response to Academic Editor’s Comments

New Editorial Comments 1: Please note that your Data Availability Statement is currently missing the DOI/accession number of each dataset OR a direct link to access each database. If your manuscript is accepted for publication, you will be asked to provide these details on a very short timeline. We therefore suggest that you provide this information now, though we will not hold up the peer review process if you are unable.

Response 1: We thank the editorial office for highlighting this important requirement regarding data accessibility. The data used in our study were obtained from the 2021 wave of the Chinese General Social Survey (CGSS), hosted by the National Survey Research Center at Renmin University of China. The dataset is accessible through the Chinese National Survey Data Archive (CNSDA). We have now updated the Data Availability Statement both at the end of the manuscript and in a separate supplementary file to include the direct access link to the dataset, as follows:

http://www.cnsda.org/index.php?r=projects/view&id=65635422

This link leads to the official data application page, where users may register, describe their research project, and submit a request for access. As stated, the authors did not receive any special privileges during data acquisition. We hope this revision meets the journal’s expectations for transparency and data availability. Please feel free to let us know if any further clarification is required.

New Editorial Comments 2: Please ensure that you refer to Figure 1 in your text as, if accepted, production will need this reference to link the reader to the figure.

Response 2: We appreciate this helpful reminder. In response, we have added a direct reference to Figure 1 in the section titled “Research Objectives and Hypotheses”, specifically in line 144 of the revised manuscript. The newly added sentence reads:

“Figure 1 presents the proposed structural model, which guides the formulation of the research hypotheses outlined below.”

This change ensures that Figure 1 is properly linked to the main text and can be correctly processed during production. Please let us know if any further clarification is needed.

Comments 1: When submitting your revision, we need you to address these additional requirements. Please ensure that your manuscript meets PLOS ONE's style requirements, including those for file naming. The PLOS ONE style templates can be found at https://journals.plos.org/plosone/s/file?id=wjVg/PLOSOne_formatting_sample_main_body.pdf and https://journals.plos.org/plosone/s/file?id=ba62/PLOSOne_formatting_sample_title_authors_affiliations.pdf

Response 1: We appreciate your kind reminder regarding PLOS ONE’s formatting standards. In response, we have thoroughly reviewed and implemented the journal’s style requirements. Specifically, we have:

1. Formatted the main manuscript according to the PLOS ONE main body template, including proper section headings, figure/table callouts, and structure.

2. Ensured that the title page and author affiliation section are presented in accordance with the designated template.

3. Updated the file naming conventions to comply with PLOS ONE’s submission policies (e.g., “Manuscript.docx” was renamed accordingly before upload).

We hope these revisions meet the journal’s formatting expectations. Please kindly let us know if any further adjustments are needed.

Comments 2: Please provide a complete Data Availability Statement in the submission form, ensuring you include all necessary access information or a reason for why you are unable to make your data freely accessible. If your research concerns only data provided within your submission, please write "All data are in the manuscript and/or supporting information files" as your Data Availability Statement.

Response 2: Thank you for your comment on the Data Availability Statement. We confirm that the data used in this study come from the 2021 Chinese General Social Survey (CGSS), which is managed by the National Survey Research Center (NSRC) at Renmin University of China. As the data are not publicly downloadable, researchers must apply for access via the CGSS official website (http://cgss.ruc.edu.cn/), and no special access privileges were granted to the authors.

We have clarified this information in the Data Availability Statement section of our manuscript and ensured that it aligns with PLOS ONE’s data sharing policy.

Comments 3: First, since your study focuses on the mediating roles of mental health-related variables: psychological anxiety, social loneliness, and goal deficiency in the relationship between internet use and well-being, it would be beneficial to elaborate on the mediating effects of each variable rather than discussing mental health in general. As Reviewer 2 pointed out, the literature review needs to be significantly strengthened. Additionally, the rationale for conceptualizing mental health as encompassing these three specific dimensions should be well-justified with theoretical support.

Response 3: We appreciate your valuable suggestions. The manuscript has been revised as follows to address your points: Expanded Mediation Analysis: In both the Results and Discussion sections, we have separately elaborated on the mediating effects of each psychological variable—psychological anxiety, social loneliness, and goal deficiency—rather than treating mental health as a single general construct. For example, we highlighted that the path via psychological anxiety showed the strongest indirect effect, while the effect of goal deficiency was weaker but still significant, and the effect of social loneliness was not statistically significant.

1. Enhanced Literature Review: In the revised Literature Review section, we have substantially expanded the discussion to include recent (post-2020) studies on internet use and mental health among older adults. These references better reflect the current state of the field and provide stronger empirical foundations for our research hypotheses (H1–H5).

2. Theoretical Justification: We have added a new paragraph to explain the theoretical basis for selecting the three psychological dimensions. Specifically: Psychological anxiety is grounded in the stress-coping theory; Social loneliness draws on social support theory; Goal deficiency is supported by motivational theory and meaning-making frameworks.

This addition appears in both the Literature Review and Method sections (specifically under the “Mediating Variables” subsection), with a clear operationalization of each construct based on validated items from the CGSS dataset.

We hope these revisions provide a clearer and more rigorous foundation for our conceptual model and empirical analyses.

Comments 4: Second, Tables 2–5 consistently show positive relationships among key variables. For example, in Table 3, the coefficient of 0.193 indicates that higher psychological anxiety was positively associated with subjective well-being—an unexpected finding that contradicts existing literature. If taken at face value, this result would suggest that increasing older adults’ anxiety levels could enhance their well-being, which is counterintuitive. Additionally, the internet use was positively associated with three mediators (i.e., psychological anxiety, social loneliness, and goal deficiency) as shown in Table 4. However, the discussion on page 23 interprets the findings contradictorily: “This finding aligns with the perspective of social compensation theory [36], suggesting that social media plays an especially prominent role in alleviating psychological anxiety.” Similarly, although goal deficiency was found to be positively related to subjective well-being in Table 3, the discussion on page 22 states: “Similarly, the reduction in goal deficiency also significantly mediated the relationship between social media use and subjective well-being. Social media provides crucial support for older adults in rebuilding their sense of purpose by offering opportunities to participate in community activities, express themselves, and learn new skills.” These interpretations seem inconsistent with the reported results. Please carefully review your data analysis procedures and ensure that the findings are precise and accurately reflected in the discussion.

Response 4: We greatly appreciate your careful attention to the consistency between the empirical results and theoretical interpretations. We sincerely apologize for the earlier inconsistency between the regression findings and the theoretical discussion, especially regarding psychological anxiety and goal deficiency. After carefully reviewing the regression results and indirect effect pathways, we have revised the relevant sections in the Discussion and Results parts to ensure consistency and accuracy in interpretation:

1. Psychological Anxiety (Table 3): The unstandardized coefficient of psychological anxiety in Table 3 is –0.193, not +0.193. We reviewed the manuscript and confirmed that this may have resulted from a typographical oversight in the previous draft. The correct interpretation — that higher anxiety is associated with lower subjective well-being — is now consistently reflected in the discussion. We have added clarification in the first and second paragraphs of the discussion to align with both Table 3 and the indirect effect shown in Table 4.

2. Goal Deficiency: Table 3 also shows a negative relationship between goal deficiency and subjective well-being. The mediation effect presented in Table 4 further supports this indirect pathway. In the earlier version, we did not fully elaborate on the mechanism. We have now refined our explanation to clearly state that a reduction in goal deficiency contributes to improved well-being, and we have revised this in the third paragraph of the discussion section.

3. Social Loneliness: While social loneliness was not a significant mediator in the indirect path model, we acknowledge that the previous discussion may have overstated its impact. We have revised the wording to emphasize that its mediating effect was not statistically significant, consistent with Table 4. This correction is included in the third paragraph of the revised discussion.

4. Theoretical Framing: To enhance conceptual clarity and ensure theoretical alignment, we have reinforced the interpretation using flow theory and social compensation theory. The causal logic linking internet use, psychological mediators, and subjective well-being is now clearly stated. All revised interpretations are now directly grounded in the empirical findings from Tables 3 and 4.

Comments 5: Furthermore, regarding Table 5, the direct effect between internet use and subjective well-being is reported as 0.025, yet the total effect (direct + indirect) is the same as the total indirect effect (0.024). Please verify these statistics. Last, please report the procedures of getting the indirect effects contrast in Table 5.

Response 5: We appreciate your close scrutiny of the reported model estimates and your attention to statistical detail. We sincerely apologize for the confusion caused by the inconsistent effect values reported in the previous version. In response, we have taken the following corrective actions:

1. We re-estimated the structural equation model using the R lavaan package, which allowed us to compute all path coefficients and mediation effects within a consistent framework. The previous inconsistency was due to a typographical oversight during table construction, which has now been corrected.

2. The revised Table 4 now presents standardized estimates for the direct, indirect, and total effects, as well as z-values, p-values, and 95% confidence intervals. The total effect of Internet Use on Subjective Well-being is now correctly reported as 0.066, which matches the sum of the direct effect (0.059) and the total indirect effect (0.029). This correction ensures statistical coherence and transparency.

3. Regarding the procedure for estimating indirect effects, we used bootstrapping with 5,000 samples to calculate bias-corrected confidence intervals for each indirect pathway. This approach is implemented via the sem() function and parameterEstimates() output in the lavaan package, which directly provides the standardized estimates, z-values, p-values, and confidence intervals used in Table 4.

We hope this resolves the issue and greatly appreciate your careful reading and valuable feedback.

Response to Reviewer 1’s Comments:

Comments 1: Statistical analyses were performed using SPSS and the PROCESS macro. I suggest using path analysis in software such as R, LISREL, or AMOS. In this way, fit indices, which are very important in modeling, are also reported.

Response 1: Building upon your insightful suggestion, we have implemented structural equation modeling (SEM) using the lavaan package in R to enhance analytical robustness and clarity. This approach enabled us to simultaneously estimate direct, indirect, and total effects, and importantly, to include model fit indices.

In the revised version of our manuscript, we have re-conducted the mediation analysis using the lavaan package in R. The SEM approach enabled us to simultaneously estimate direct, indirect, and total effects, and more importantly, to report a set of model fit indices. As a result, we have:

1. Reconstructed Table 4 to present standardized path coefficients, z-values, p-values, and confidence intervals for all key paths.

2. Added Table 5, which reports model fit indices including χ²/df, RMSEA, GFI, AGFI, NFI, NNFI, and IFI, all of which indicate acceptable model fit according to conventional criteria.

We believe that this change has greatly improved the transparency and methodological rigor of our analysis. The corresponding descriptions in the Results section have also been updated accordingly.

Comments 2: Assumptions of statistical analysis should be checked and reported.

Response 2: We fully recognize the importance of verifying and transparently reporting the assumptions underlying statistical analysis. In response to your suggestion, we carefully re-examined our analytical procedures and have added explicit clarification in the Statistical Analysis subsection of the Methods section. Specifically, we now state that essential assumptions—such as normality, linearity, and multicollinearity—were assessed prior to conducting the structural equation modeling (SEM) using the lavaan package in R. Additionally, model fit indices (e.g., RMSEA, GFI, NFI, CFI) are now reported in the newly added Table 5, in alignment with conventional SEM reporting standards. We sincerely hope that these improvements address your concern and enhance the methodological rigor and transparency of the study. Thank you again for your constructive feedback.

Comments 3: Another advantage of using path analysis in the way I mentioned is that all hypotheses are tested simultaneously, and a path diagram is also reported. Your results are fragmented.

Response 3: We fully concur with your observation that structural path modeling provides a more integrated and interpretable framework for testing all hypothesized relationships simultaneously. We also agree t

---

## [Decision Letter · Decision Letter 1]

20 Jun 2025

Dear Dr. Tan,

Thank you for submitting your manuscript to PLOS ONE. After careful consideration, we feel that it has merit but does not fully meet PLOS ONE’s publication criteria as it currently stands. Therefore, we invite you to submit a revised version of the manuscript that addresses the points raised during the review process.

We look forward to receiving your revised manuscript.

Kind regards,

Lianshan Zhang

Academic Editor

PLOS ONE

Reviewers' comments:

Reviewer's Responses to Questions

**Comments to the Author**

Reviewer #2: All comments have been addressed

Reviewer #3: All comments have been addressed

2. Is the manuscript technically sound, and do the data support the conclusions?

Reviewer #2: Yes

Reviewer #3: Partly

3. Has the statistical analysis been performed appropriately and rigorously?

Reviewer #2: Yes

Reviewer #3: Yes

4. Have the authors made all data underlying the findings in their manuscript fully available?

Reviewer #2: Yes

Reviewer #3: Yes

5. Is the manuscript presented in an intelligible fashion and written in standard English?

Reviewer #2: Yes

Reviewer #3: Yes

Reviewer #2: (No Response)

Reviewer #3: Comment 1:

In the first paragraph of the introduction, you stated: “Against this backdrop, the widespread adoption of the internet is gradually becoming a key pathway to enhancing the subjective well-being of older adults. Exploring its underlying mechanisms holds significant theoretical value and practical implications.” (lines 46–49). This statement appears somewhat arbitrary, as the existing literature provides contradictory evidence on whether internet use improves or reduces users’ well-being. Your study focuses solely on the positive effects of internet use on subjective well-being. Please provide stronger justification for your research aim and situate it within the broader context of internet use literature. Referring to works such as Çikrıkci (2016) and Heo et al. (2015) may help strengthen your argument:

Çikrıkci, Ö. (2016). The effect of internet use on well-being: Meta-analysis. Computers in Human Behavior, 65, 560–566. https://doi.org/10.1016/j.chb.2016.09.021

Heo, J., Chun, S., Lee, S., Lee, K. H., & Kim, J. (2015). Internet use and well-being in older adults. Cyberpsychology, Behavior, and Social Networking, 18(5), 268–272.

Comment 2:

The third paragraph of the introduction attempts to justify why you focus on mental health as a potential mechanism underlying the relationship between internet use and subjective well-being. However, the current arguments, such as moderate or excessive internet use and the social compensation theory, focus more on how people use the internet and which kinds of people benefit in terms of well-being. These points address the boundary conditions of the relationship between internet use and subjective well-being, rather than justifying the inclusion of mental health as a mediator. A more logical and robust argument is needed to justify why mental health is the focus of your study.

Comment 3:

There is a need to identify gaps in existing studies on the relationship between internet use and well-being to justify the theoretical implications of your research. Introducing such research gaps would help strengthen the introduction and highlight the relevance of this study.

Comment 4:

In the first section of the literature review, you wrote: “Despite these insights, existing research has primarily emphasized the outcomes of internet use on well-being, while neglecting the internal psychological processes through which such effects occur. Specifically, the role of psychological states—such as anxiety, loneliness, and goal perception—has received limited attention as potential mediators.” This statement suggests that your selection of these mediators was somewhat arbitrary and lacks theoretical justification. Please explain why you focus on psychological states and, specifically, why you chose anxiety, loneliness, and goal perception as mediators.

Comment 5:

Throughout the introduction and literature review, you refer to “internet use,” while in the hypotheses section, you focus on “social media use.” These are distinct concepts, and the inconsistency may confuse readers. Please use consistent terminology throughout the manuscript to avoid misunderstandings.

Comment 6:

Each hypothesis lacks sufficient elaboration, making the current literature review appear weak. At least three references should be provided to justify each hypothesis. For example, you did not cite any literature to support the hypothesized relationship between internet use, goal deficiency, and subjective well-being. Please include relevant references to strengthen the justification for your hypotheses.

Comment 7:

In the introduction and literature review, please explain why your study focuses on older adults. Are there unique patterns of internet use within this population? Does internet use have distinctive impacts on their mental health or well-being? More elaboration on the research context is needed to clarify the rationale behind focusing on older adults.

Comment 8:

In the “Statistical Analysis” section, you mentioned: “Building on this, mediation effects were tested to explore the mechanisms among internet use, alienation, and mental health. The mediating role of alienation in the relationship between internet use and mental health was examined.” However, “alienation” does not appear in the results. Why is this variable mentioned here but not included in the findings? Additionally, why did you not include a structural equation modeling (SEM) approach to test mediation effects in this section?

Comment 9:

When reporting the correlation analysis in the results, you wrote: “These results indicate that subjective well-being among older adults is influenced by internet use, education, and psychological status. Thus, promoting positive internet engagement, expanding educational access, and strengthening mental health support may be beneficial to the well-being of this population.”

However, as this is a cross-sectional survey, causal relationships between variables cannot be inferred. Please avoid using the term “influence” when reporting correlation results. Additionally, the suggested implications (e.g., promoting positive internet engagement) should be moved to the discussion section, as they do not belong in the results.

Comment 10:

In the section “The Relationship Between Internet Use and Subjective Well-Being Among Older Adults,” there are inconsistencies in how standard errors (SE) are reported. For some correlations, SE is provided, while for others, it is not. Please ensure consistent reporting of results and clarify the reasons for these discrepancies, if any.

Comment 11:

What does “-Cons” mean in Table 3? Please clarify or provide a footnote to explain this abbreviation.

Comment 12:

In Table 4, the total effect of internet use on subjective well-being is reported as 0.066. However, this value does not equal the sum of the direct effect (0.059) and the total indirect effect (0.029). Please provide an explanation for this discrepancy.

Comment 13:

The title of Table 5 reads: “Table 5. Pearson’s correlations among relevant study variables.” However, the table seems to pertain to the model fit of SEM. Additionally, you did not report SRMR and CFI values for the model, which are essential for assessing model fit. Please revise this section and refer to Hu and Bentler (1999) for appropriate model fit criteria:

Hu, L., & Bentler, P. M. (1999). Cutoff criteria for fit indexes in covariance structure analysis: Conventional criteria versus new alternatives. Structural Equation Modeling, 6(1), 1–55. https://doi.org/10.1080/10705519909540118

Comment 14:

In the discussion section, avoid repeating the findings already reported in the results. Instead, compare your findings with those of existing studies, provide plausible explanations, and suggest implications. Please remove statistical details from the discussion section and focus on strengthening the theoretical implications of your findings.

Comment 15:

Please include a discussion of the study’s limitations, such as the measurement of key variables, to clarify the generalizability of your findings. Addressing limitations will enhance the transparency and rigor of your study.

**Do you want your identity to be public for this peer review?** For information about this choice, including consent withdrawal, please see our Privacy Policy

Reviewer #2: **Yes: ** ZHANG Lin

Reviewer #3: No

---

## [Author Response · Author response to Decision Letter 2]

30 Jun 2025

Dear Academic Editor and Reviewers,

We would like to express our sincere gratitude to the academic editor and all reviewers for your thorough evaluation and constructive feedback on our manuscript. We greatly appreciate the time and effort you invested in helping us improve the clarity, conceptual grounding, and methodological rigor of this study.

In response to each comment, we have carefully revised the manuscript and provided point-by-point answers below. For clarity, each reviewer comment is restated, followed by our detailed response and the specific location of the corresponding changes in the revised manuscript.

Response to Reviewer 3’s Comments:

Comments 1: In the first paragraph of the introduction, you stated: “Against this backdrop, the widespread adoption of the internet is gradually becoming a key pathway to enhancing the subjective well-being of older adults. Exploring its underlying mechanisms holds significant theoretical value and practical implications.” (lines 46–49). This statement appears somewhat arbitrary, as the existing literature provides contradictory evidence on whether internet use improves or reduces users’ well-being. Your study focuses solely on the positive effects of internet use on subjective well-being. Please provide stronger justification for your research aim and situate it within the broader context of internet use literature. Referring to works such as Çikrıkci (2016) and Heo et al. (2015) may help strengthen your argument:

Çikrıkci, Ö. (2016). The effect of internet use on well-being: Meta-analysis. Computers in Human Behavior, 65, 560–566. https://doi.org/10.1016/j.chb.2016.09.021

Heo, J., Chun, S., Lee, S., Lee, K. H., & Kim, J. (2015). Internet use and well-being in older adults. Cyberpsychology, Behavior, and Social Networking, 18(5), 268–272.

Response 1: Thank you for this valuable comment. We agree that the original statement in the introduction could benefit from a more nuanced and balanced perspective. In the revised manuscript�we have incorporated a broader view of the literature, including studies that report both positive and more critical findings on the relationship between internet use and subjective well-being. Specifically, we have referred to Çikrıkci (2016) and Heo et al. (2015), whose work underscores the complexity and variability of this relationship. These revisions aim to provide stronger justification for our research aim and highlight the theoretical relevance of investigating the underlying mechanisms through which internet use may affect older adults’ well-being.

Manuscript revision location: Introduction, lines 52–57.

Comment 2: The third paragraph of the introduction attempts to justify why you focus on mental health as a potential mechanism underlying the relationship between internet use and subjective well-being. However, the current arguments, such as moderate or excessive internet use and the social compensation theory, focus more on how people use the internet and which kinds of people benefit in terms of well-being. These points address the boundary conditions of the relationship between internet use and subjective well-being, rather than justifying the inclusion of mental health as a mediator. A more logical and robust argument is needed to justify why mental health is the focus of your study.

Response 2: We appreciate the reviewer’s insightful comment. In response, we have revised the third paragraph of the introduction to provide a more theoretically grounded and logically cohesive justification for selecting mental health as the mediating variable. Rather than focusing solely on usage patterns or population differences, we now draw on psychological pathway mechanisms and relevant theories such as Socioemotional Selectivity Theory and Life Course Theory. These frameworks emphasize how age-related psychosocial transitions influence emotional regulation and mental adaptation, thereby highlighting mental health as a core mechanism through which internet use may affect subjective well-being among older adults.

We have incorporated this rationale to better align with the mediating focus of our study and added references to support our argument. These changes clarify the theoretical contribution of the mental health pathway and address the concern regarding its conceptual justification.

Manuscript revision location: Introduction, lines 68–86.

Comment 3: There is a need to identify gaps in existing studies on the relationship between internet use and well-being to justify the theoretical implications of your research. Introducing such research gaps would help strengthen the introduction and highlight the relevance of this study.

Response 3: Thank you for this insightful comment. We agree that identifying the research gaps is crucial to demonstrating the theoretical relevance of our study. In the revised introduction, we have added a paragraph that explicitly highlights the limitations of existing research, particularly the lack of attention to psychological mechanisms linking internet use and subjective well-being among older adults. While prior studies have often focused on direct associations, few have systematically examined how mental health functions as a mediating pathway. By emphasizing this gap, we aim to clarify the theoretical contribution of our study and provide a stronger rationale for exploring mental health as a core mechanism in this relationship.

Manuscript revision location: Introduction, lines 87–98

Comment 4: In the first section of the literature review, you wrote: “Despite these insights, existing research has primarily emphasized the outcomes of internet use on well-being, while neglecting the internal psychological processes through which such effects occur. Specifically, the role of psychological states—such as anxiety, loneliness, and goal perception—has received limited attention as potential mediators.” This statement suggests that your selection of these mediators was somewhat arbitrary and lacks theoretical justification. Please explain why you focus on psychological states and, specifically, why you chose anxiety, loneliness, and goal perception as mediators.

Response 4: We sincerely appreciate the reviewer’s insightful comment regarding the justification of our chosen mediating variables. In response, we have added a more detailed theoretical explanation to support the inclusion of psychological anxiety, social loneliness, and goal deficiency as mediators in the relationship between internet use and subjective well-being.

Firstly, psychological anxiety is a fundamental emotional response to perceived uncertainty or stress, which is highly relevant in the digital context. According to Pearlin’s stress process model [24], exposure to complex digital information or excessive online stimulation may intensify psychological pressure, especially among older adults with lower digital literacy. Thus, internet use may influence well-being by either alleviating or exacerbating anxiety, depending on usage patterns.

Secondly, social loneliness represents a perceived lack of meaningful social ties. As Weiss [26] proposed in his theory of relational loneliness, older individuals are especially prone to feelings of isolation due to reduced social roles in later life. The internet, while offering communication tools, may not always substitute for authentic offline relationships. Therefore, loneliness serves as a critical pathway through which internet use affects emotional well-being.

Thirdly, goal deficiency is linked to an individual’s sense of direction and purpose in life. Drawing from Self-Determination Theory by Deci and Ryan [25], individuals require autonomy and competence to sustain psychological wellness. When older adults engage in internet use without purposeful engagement—such as passive scrolling—they may experience a diminished sense of personal achievement, contributing to goal deficiency. Conversely, when online activities align with intrinsic motivations, the internet can help strengthen one’s life purpose.

These three constructs are not arbitrarily selected but are grounded in well-established psychological theories. They represent emotional (anxiety), relational (loneliness), and motivational (goal perception) domains of mental health, thus offering a comprehensive framework to understand how digital engagement translates into subjective well-being outcomes. We have incorporated these theoretical foundations into the “Internet Use and Mental Health” and “Mental Health and Subjective Well-Being” sections of the literature review to clarify our rationale.

Manuscript revision location: Literature review, lines 99–164.

Comment 5: Throughout the introduction and literature review, you refer to “internet use,” while in the hypotheses section, you focus on “social media use.” These are distinct concepts, and the inconsistency may confuse readers. Please use consistent terminology throughout the manuscript to avoid misunderstandings.

Response 5: We sincerely thank the reviewer for pointing out the inconsistency in terminology between “internet use” and “social media use.” We fully acknowledge that these terms refer to distinct concepts, and such inconsistency could lead to confusion regarding the study’s focus and scope.

To address this issue, we have carefully reviewed the entire manuscript and standardized the terminology to consistently use “internet use” throughout, including in the hypotheses section, methods, and discussion. All references to “social media use” have been revised accordingly to ensure conceptual clarity and alignment with our research framework.

We appreciate the reviewer’s careful attention to detail, which has helped improve the coherence and rigor of the manuscript.

Manuscript revision location: Throughout the manuscript, including the Hypotheses section (Lines 178–203).

Comment 6: Each hypothesis lacks sufficient elaboration, making the current literature review appear weak. At least three references should be provided to justify each hypothesis. For example, you did not cite any literature to support the hypothesized relationship between internet use, goal deficiency, and subjective well-being. Please include relevant references to strengthen the justification for your hypotheses.

Response 6: Thank you very much for this valuable suggestion. We agree that the initial version of the manuscript lacked adequate theoretical elaboration and supporting references for the stated hypotheses, particularly regarding the link between internet use, goal deficiency, and subjective well-being.

In response, we have revised the Research Hypotheses section to include more detailed theoretical justifications for each hypothesis. For each proposed pathway, we have now incorporated at least three relevant, high-quality references to support the rationale, drawing upon recent literature in aging, mental health, and digital behavior. Specifically, for H4, we added references such as Zhang and Zhou (2022), Sum et al. (2008), and Nie et al. (2023), which empirically demonstrate that excessive or unstructured internet use may reduce older adults’ goal clarity and perceived life purpose, thereby negatively impacting subjective well-being.

These modifications not only reinforce the theoretical foundation of our study but also help clarify the conceptual pathways being tested. We hope these improvements will enhance the scholarly rigor and clarity of the manuscript.

Manuscript Revision Location: Research Objectives and Hypotheses, Lines 178–203 (Theoretical justifications and references have been added before each hypothesis).

Comment 7: In the introduction and literature review, please explain why your study focuses on older adults. Are there unique patterns of internet use within this population? Does internet use have distinctive impacts on their mental health or well-being? More elaboration on the research context is needed to clarify the rationale behind focusing on older adults.

Response 7: We thank the reviewer for this insightful comment. In the revised version of the manuscript, we have strengthened the rationale for focusing on older adults by clarifying the unique characteristics that make this population especially relevant to the research question.

First, we emphasized that older adults exhibit distinct internet use patterns, often using digital platforms for emotionally meaningful goals—such as staying in touch with family, seeking health-related information, and alleviating loneliness—which differ substantially from younger users’ motivations (e.g., entertainment or work).

Second, older adults are psychologically more vulnerable due to age-related life transitions (e.g., retirement, bereavement, health decline), and their subjective well-being tends to be more sensitive to emotional states and social connectivity. Research has shown that this group faces a higher risk of mental health challenges, including loneliness, anxiety, and a reduced sense of purpose, making them an ideal population for studying how internet use may help or hinder well-being through psychological pathways.

Third, given the rapid digital inclusion of older adults in countries like China, this group represents both a policy and scientific priority for understanding how to promote healthy aging in the digital age.

Manuscript revision location: Literature review, lines 58–67.

Comment 8: In the “Statistical Analysis” section, you mentioned: “Building on this, mediation effects were tested to explore the mechanisms among internet use, alienation, and mental health. The mediating role of alienation in the relationship between internet use and mental health was examined.” However, “alienation” does not appear in the results. Why is this variable mentioned here but not included in the findings? Additionally, why did you not include a structural equation modeling (SEM) approach to test mediation effects in this section?

Response 8: Thank you for pointing this out. We acknowledge that the mention of “alienation” in the original statistical analysis section was an oversight. The term was used in an earlier draft of the manuscript during exploratory discussions of potential mediators but was later removed from the analysis due to insufficient theoretical fit and measurement limitations. We have now deleted the reference to “alienation” in the Statistical Analysis section to maintain consistency with the final reported results.

In response to your second point, we confirm that we did use structural equation modeling (SEM) to test the mediating effects of psychological anxiety, social loneliness, and goal deficiency between internet use and subjective well-being. The analysis was conducted using the lavaan package in R, and the results are presented in Table 4 of the revised manuscript. The significance of indirect effects was assessed using the bias-corrected bootstrap method with 5,000 resamples. We have revised the Statistical Analysis section accordingly to clarify this methodological approach.

Manuscript changes: Statistical Analysis section, lines 275-285.

Comment 9: When reporting the correlation analysis in the results, you wrote: “These results indicate that subjective well-being among older adults is influenced by internet use, education, and psychological status. Thus, promoting positive internet engagement, expanding educational access, and strengthening mental health support may be beneficial to the well-being of this population.”

However, as this is a cross-sectional survey, causal relationships between variables cannot be inferred. Please avoid using the term “influence” when reporting correlation results. Additionally, the suggested implications (e.g., promoting positive internet engagement) should be moved to the discussion section, as they do not belong in the results.

Response 9: Thank you for your insightful comment. We agree that using the term “influence” in the context of correlation analysis within a cross-sectional study may lead to misinterpretation of causality. We have revised the wording in the Results section to use neutral expressions such as “associated with” and “correlated with,” avoiding any implication of causality. Additionally, we have removed the recommendation-oriented sentences (e.g., “promoting positive i

---

## [Decision Letter · Decision Letter 2]

12 Aug 2025

Dear Dr. Tan,

Thank you for submitting your manuscript to PLOS ONE. After careful consideration, we feel that it has merit but does not fully meet PLOS ONE’s publication criteria as it currently stands. Therefore, we invite you to submit a revised version of the manuscript that addresses the points raised during the review process.

We look forward to receiving your revised manuscript.

Kind regards,

Lianshan Zhang

Academic Editor

PLOS ONE

Journal Requirements:

Reviewers' comments:

Reviewer's Responses to Questions

**Comments to the Author**

Reviewer #3: (No Response)

2. Is the manuscript technically sound, and do the data support the conclusions?

Reviewer #3: Partly

3. Has the statistical analysis been performed appropriately and rigorously?

Reviewer #3: Yes

4. Have the authors made all data underlying the findings in their manuscript fully available?

Reviewer #3: Yes

5. Is the manuscript presented in an intelligible fashion and written in standard English?

Reviewer #3: Yes

Reviewer #3: The current manuscript has improved significantly in terms of its context, research gaps in existing literature, and justification for examining the mediating effect of three mental health indicators. However, I still have some comments regarding the literature review and statistics reporting.

1.Citations Needed:

a.The content about the uniqueness of older adults from lines 61 to 67 needs citations:

“Older adults exhibit unique patterns of internet use, often centered around maintaining family contact, acquiring health-related information, or alleviating loneliness. At the same time, they are more vulnerable to mental health challenges due to life course transitions, reduced social networks, and cognitive aging. Compared to younger cohorts, older adults’ subjective well-being is more sensitive to changes in emotional states and social connectedness. These characteristics make them a critical group for examining how internet use influences well-being through psychological mechanisms.”

b.Similarly, “Older adults are especially sensitive to psychological vulnerabilities, as they often experience transitions such as retirement, bereavement, and social role reduction, which may affect their emotional and cognitive well-being.” (Lines 71-73) needs citation.

c.“Despite growing scholarly interest in how internet use affects the subjective well-being of older adults, existing studies have primarily concentrated on its direct effects, with insufficient attention to the underlying psychological mechanisms. In particular, the mediating role of mental health remains underexplored. Although previous research has shown associations between internet use and emotional outcomes, few studies have systematically investigated how specific psychological constructs—such as anxiety, loneliness, or goal-related cognition—function as mediators in this relationship. Addressing this gap is essential for advancing theoretical understanding and informing targeted mental health interventions in digitally connected aging populations.” (Lines 87-95) also needs citation.

d.“For instance, older adults experiencing lower anxiety and loneliness are more likely to benefit from online interactions, while those who lack a sense of life purpose may not derive meaningful satisfaction from internet use.” (Lines 157-159) needs citation.

2.Theoretical Arguments:

a.“According to Pearlin’s stress process model [24], exposure to complex digital information or excessive online stimulation may intensify psychological pressure, especially among older adults with lower digital literacy.” However, Pearlin’s stress process model did not suggest the premise that you propose—that online information can be a source of stress. Please check this citation to avoid mis-citation.

In your literature, you also mentioned, “On the other hand, excessive or unregulated internet use may lead to information overload, social comparison, and reduced real-life interaction, which can heighten psychological anxiety and emotional exhaustion [15]. This phenomenon can be interpreted through the Stress Process Model [24], which posits that environmental demands—such as digital complexity—may trigger anxiety unless appropriate coping resources are in place.” (Lines 127-132). I don’t agree with this argument that internet use turns out to be an environmental demand that triggers people’s anxiety. You may consider interpreting the negative impacts of internet use on mental health from other theoretical perspectives.

b.Additionally, you acknowledge that “internet use may influence well-being by either alleviating or exacerbating anxiety, depending on usage patterns.” Internet usage patterns are important, but this study did not examine them, thus making the rationale for the hypothesis quite weak.

3.Literature Review:

a.In the literature review “Internet Use and Mental Health,” you concluded, “Taken together, these findings suggest that the effects of internet use on mental health are not linear, but mediated by key psychological variables.” However, according to your theoretical arguments, it seems that the relationships between internet use and different aspects of mental health mainly depend on Internet usage patterns, such as excessive or unregulated internet use, and digital engagement that lacks meaningful goal orientation. However, your study did not measure older adults’ usage patterns; instead, you only measured their frequency of internet use. Thus, the current theoretical argument seems not to provide a strong rationale for your research focus and hypothesis.

b.In your response to Comment 4, you mentioned why you chose anxiety, loneliness, and goal perception as mediators: “These three constructs are not arbitrarily selected but are grounded in well-established psychological theories. They represent emotional (anxiety), relational (loneliness), and motivational (goal perception) domains of mental health, thus offering a comprehensive framework to understand how digital engagement translates into subjective well-being outcomes.” This argument looks reasonable, but why don’t you justify it in the introduction and literature review?

4.Research Objective & Hypotheses:

a.The research objective, “The aim of this study is to examine the relationships among internet use, mental health, and subjective well-being (SWB) in older adults, with a particular focus on the mediating role of mental health. Although internet use among older adults is increasingly prevalent, the psychological mechanisms linking internet use to well-being remain insufficiently understood. Drawing on nationally representative data from the China General Social Survey (CGSS), this study investigates how internet use influences SWB through three specific psychological dimensions: psychological anxiety, social loneliness, and goal deficiency,” should be moved up to the end of the introduction instead of being positioned behind the literature review.

b.The hypotheses section should also be inserted into the literature review when you state the relationship between key variables. The current formatting is redundant and repetitive.

c.H5 is actually the composition of H2-4. Why bother to include it? Additionally, the relationships between internet use and the three mental health dimensions are missing in the current hypotheses.

An alternative way is to propose hypotheses on the relationship between internet use and the three mental health dimensions and then keep the current H5 (i.e., the mediation effect of the three mental health dimensions in the relationship between internet use and subjective well-being). This can also help readers understand the direction and valence of each path.

There are many tutorials about how to propose direct and mediating hypotheses in research. You may consider referring to them.

5.Measurement

a.“Variable manipulation” is often used in experiments when you manipulate a variable through stimuli. Please change it to “variable” or just remove it.

b.Your conceptualization of the three dimensions of mental health and justification should be incorporated into the literature review instead of being presented in the operationalization of measurement.

c.In measurement, the symbol of mean (M) and standard deviation (SD) for each variable should be italicized. Additionally, why did you not include the values of the mean and standard deviation for other variables, such as SWB, internet use, and control variables? Please go through your manuscript closely to ensure that all statistics are reported consistently.

6.Statistical Analysis:

Please indicate how you evaluate the model fit of the SEM model, specifically, which indicators you use and which cut-off values you used for benchmarking, including citations.

7.Results:

a.In descriptive statistics, “Regarding place of residence, 94.3% of the participants lived in rural areas, while only 5.7% were from urban areas.” Is this true? Why does the majority of older adults in the sample come from rural areas instead of urban areas, which is quite counterintuitive? Isn’t internet access among older adults in urban areas higher than those in rural areas? Given this sample characteristic, the findings of this study have limitations in generalizability. Please acknowledge this in the limitations.

b.The symbol of correlation and p-value should also be italicized.

c.When reporting the results of “Model 3: Inclusion of Psychological Anxiety,” “of birth (B = –0.013, p < 0.05) and education (B = 0.092, p < 0.05) remain significant; gender and residence remain insignificant.” “Insignificant” is the wrong expression; please replace it with “non-significant.”

d.The font of “Standard Error” in Table 4 is different from other values. Please keep all fonts of values consistent.

e.When reporting the results in “Mediating Effect Analysis,” you should always address the hypotheses you proposed in the literature review to show readers which ones are supported and which ones are rejected.

8.Discussion:

a.You explained the positive association between internet and subjective well-being. However, the results of the Stepwise OLS Regression Analysis and SEM both suggest these two variables have almost non-significant correlation. Given this, how can you still draw such a conclusion? Doesn’t the non-significant association between internet use and subjective well-being suggest that internet use itself cannot predict subjective well-being? Instead, usage patterns (i.e., how they use the internet) matter in this process.

9.What is missing in the current discussion is the practical implications of the findings. How do your findings about the relationships between internet use, mental health, and subjective well-being provide insights for older adults and institutions to improve the well-being of this vulnerable population? Additionally, how can the findings be generalized to other contexts where older adults may have different levels of internet literacy or mental health issues?

**Do you want your identity to be public for this peer review?** For information about this choice, including consent withdrawal, please see our Privacy Policy

Reviewer #3: No

---

## [Author Response · Author response to Decision Letter 3]

25 Aug 2025

Response to Reviewer 3’s Comments:

Comments 1: Citations Needed:

a. The content about the uniqueness of older adults from lines 61 to 67 needs citations:

“Older adults exhibit unique patterns of internet use, often centered around maintaining family contact, acquiring health-related information, or alleviating loneliness. At the same time, they are more vulnerable to mental health challenges due to life course transitions, reduced social networks, and cognitive aging. Compared to younger cohorts, older adults’ subjective well-being is more sensitive to changes in emotional states and social connectedness. These characteristics make them a critical group for examining how internet use influences well-being through psychological mechanisms.”

b. Similarly, “Older adults are especially sensitive to psychological vulnerabilities, as they often experience transitions such as retirement, bereavement, and social role reduction, which may affect their emotional and cognitive well-being.” (Lines 71-73) needs citation.

c. “Despite growing scholarly interest in how internet use affects the subjective well-being of older adults, existing studies have primarily concentrated on its direct effects, with insufficient attention to the underlying psychological mechanisms. In particular, the mediating role of mental health remains underexplored. Although previous research has shown associations between internet use and emotional outcomes, few studies have systematically investigated how specific psychological constructs—such as anxiety, loneliness, or goal-related cognition—function as mediators in this relationship. Addressing this gap is essential for advancing theoretical understanding and informing targeted mental health interventions in digitally connected aging populations.” (Lines 87-95) also needs citation.

d. “For instance, older adults experiencing lower anxiety and loneliness are more likely to benefit from online interactions, while those who lack a sense of life purpose may not derive meaningful satisfaction from internet use.” (Lines 157-159) needs citation.

Response 1: We thank the reviewer for the insightful and constructive comments regarding the need for additional citations to support key statements about older adults’ internet use patterns, psychological vulnerabilities, and the mediating role of mental health. We fully agree that providing robust empirical and theoretical evidence will strengthen the credibility and academic rigor of our manuscript.

Accordingly, we have carefully reviewed recent high-quality publications indexed in SSCI/SCI journals and other authoritative sources and have supplemented the relevant sections with appropriate citations. Specifically:

A. For Comment a (Lines 61–67), we have added empirical studies such as Cotten et al. (2013) and Urbaniak et al. (2023), which document older adults’ distinctive internet use purposes (e.g., maintaining family ties, reducing loneliness) and the impact of life-course transitions on social relationships and well-being. We have also included Siedlecki et al. (2014) to highlight the association between social support and subjective well-being across age.

B. For Comment b (Lines 71–73), we have incorporated Li et al. (2022), which provides national longitudinal evidence from China showing that internet use can buffer the negative effects of social isolation on depression and cognitive function among older adults.

C. For Comment c (Lines 87–95), we have cited Chen and Persson (2002), Heo et al. (2015), and Tsai et al. (2025), which examine the relationship between internet use and psychological well-being, and identify technology anxiety, loneliness, and other psychological constructs as potential mediators in technology adoption and well-being outcomes.

D. For Comment d (Lines 157–159), we have included Hofer and Hargittai (2024), which empirically demonstrates that online social engagement is associated with lower levels of depression and anxiety among older adults, supporting our argument that reducing psychological distress can enhance the benefits of internet use.

We believe these revisions provide stronger support for the key arguments by incorporating recent, high-quality scholarly evidence, which we hope addresses the reviewer’s concerns and further strengthens the manuscript’s theoretical and empirical grounding.

Manuscript revision location: lines 81–240.

Comment 2-a: Theoretical Arguments:

a. “According to Pearlin’s stress process model [24], exposure to complex digital information or excessive online stimulation may intensify psychological pressure, especially among older adults with lower digital literacy.” However, Pearlin’s stress process model did not suggest the premise that you propose—that online information can be a source of stress. Please check this citation to avoid mis-citation.

In your literature, you also mentioned, “On the other hand, excessive or unregulated internet use may lead to information overload, social comparison, and reduced real-life interaction, which can heighten psychological anxiety and emotional exhaustion [15]. This phenomenon can be interpreted through the Stress Process Model [24], which posits that environmental demands—such as digital complexity—may trigger anxiety unless appropriate coping resources are in place.” (Lines 127-132). I don’t agree with this argument that internet use turns out to be an environmental demand that triggers people’s anxiety. You may consider interpreting the negative impacts of internet use on mental health from other theoretical perspectives.

Response 2-a: We sincerely appreciate the reviewer’s insightful comment and fully agree with the concern regarding the misapplication of Pearlin’s Stress Process Model. Upon careful re-examination of the original model, we acknowledge that it primarily addresses the role of stressors, mediators, and outcomes in the context of life stress, but it does not explicitly conceptualize online information exposure or digital complexity as environmental demands. We agree that our earlier framing risked overstating the model’s scope.

In response, we have revised the relevant section to remove the direct attribution to Pearlin’s Stress Process Model when describing the potential negative impacts of internet use. Instead, we now interpret these effects through a framework more directly aligned with technology-related stress research—drawing on the concept of digital stress as operationalized by Fischer et al. (2021), which captures stress perceptions arising from technology-related demands, including information overload, digital communication overload, and exposure to potentially misleading information. This theoretical adjustment ensures that our interpretation is supported by an established framework in the digital era and avoids mis-citation.

Manuscript revision location: Literature review, lines 192–197.

Comment 2-b:

b. Additionally, you acknowledge that “internet use may influence well-being by either alleviating or exacerbating anxiety, depending on usage patterns.” Internet usage patterns are important, but this study did not examine them, thus making the rationale for the hypothesis quite weak.

Response 2-b: We thank the reviewer for this helpful comment. In the revised manuscript, we have removed the reference to “usage patterns” to avoid implying that this study directly examined such variables. Instead, we have clarified that our analysis focuses on the overall frequency of internet use and its association with mental health outcomes.

Comment 3-a: Literature Review

a. In the literature review “Internet Use and Mental Health,” you concluded, “Taken together, these findings suggest that the effects of internet use on mental health are not linear, but mediated by key psychological variables.” However, according to your theoretical arguments, it seems that the relationships between internet use and different aspects of mental health mainly depend on Internet usage patterns, such as excessive or unregulated internet use, and digital engagement that lacks meaningful goal orientation. However, your study did not measure older adults’ usage patterns; instead, you only measured their frequency of internet use. Thus, the current theoretical argument seems not to provide a strong rationale for your research focus and hypothesis.

Response 3-a: We sincerely thank the reviewer for the helpful guidance. To better align our theoretical framing with the variables analyzed, we now ground our rationale in Self-Determination Theory, Socioemotional Selectivity Theory, and the life-course/transactional stress–coping perspective, which collectively support treating internet-use frequency as a meaningful exposure for affective and relational processes. We have removed wording that could imply reliance on unmeasured usage patterns and added frequency-focused empirical evidence (Heo et al., 2015; Cotten et al., 2013; Li et al., 2022). The revised paragraph clarifies that our mediation analysis examines how anxiety, social loneliness, and goal deficiency transmit the association from use frequency to subjective well-being in later life. We appreciate this suggestion, which has helped us tighten the logic and maintain a cautious, association-based interpretation of results.

Manuscript revision location: Literature Review, lines 198–230.

Comment 3-b:

b. In your response to Comment 4, you mentioned why you chose anxiety, loneliness, and goal perception as mediators: “These three constructs are not arbitrarily selected but are grounded in well-established psychological theories. They represent emotional (anxiety), relational (loneliness), and motivational (goal perception) domains of mental health, thus offering a comprehensive framework to understand how digital engagement translates into subjective well-being outcomes.” This argument looks reasonable, but why don’t you justify it in the introduction and literature review?

Response 3-b: Thank you for the helpful suggestion. We now justify our choice of the three mediators in the Literature Review by adding concise, theory-anchored sentences that map each mediator to a relevant framework. For example, we state:

•Anxiety: “From a stress–coping perspective, higher overall exposure to digital demands may be appraised as strain when coping resources are limited; therefore, we treat anxiety as a proximal affective pathway linking internet-use frequency to well-being [31,32].”

•Loneliness: “At the same time, age-related changes can erode social roles and network structure; as Weiss’s relational loneliness framework highlights, older adults are therefore especially susceptible to isolation, underscoring the salience of the loneliness pathway in later life [33].”

•Goal deficiency: “Grounded in Self-Determination Theory, psychological well-being depends on a sustained sense of purpose and competence; accordingly, overall internet-use frequency can shape affective and relational states that support—or undermine—these needs [34,35].”

These additions clarify why loneliness), goal deficiency, and anxiety are appropriate mediators and align the argument with our measured exposure (internet-use frequency).

Manuscript revision location: Literature Review, lines 196–203.

Comment 4-a: Research Objective & Hypotheses:

a. The research objective, “The aim of this study is to examine the relationships among internet use, mental health, and subjective well-being (SWB) in older adults, with a particular focus on the mediating role of mental health. Although internet use among older adults is increasingly prevalent, the psychological mechanisms linking internet use to well-being remain insufficiently understood. Drawing on nationally representative data from the China General Social Survey (CGSS), this study investigates how internet use influences SWB through three specific psychological dimensions: psychological anxiety, social loneliness, and goal deficiency,” should be moved up to the end of the introduction instead of being positioned behind the literature review.

Response 4-a: Thank you for the valuable suggestion. We have repositioned the research objective paragraph to the end of the Introduction to neatly close the motivation and research gap, enhance narrative coherence, and provide a clear bridge to the subsequent hypotheses and analytic framework.

Manuscript revision location: Introduction, lines 141–147.

Comment 4-b, c:

b. The hypotheses section should also be inserted into the literature review when you state the relationship between key variables. The current formatting is redundant and repetitive.

c.H5 is actually the composition of H2-4. Why bother to include it? Additionally, the relationships between internet use and the three mental health dimensions are missing in the current hypotheses.

An alternative way is to propose hypotheses on the relationship between internet use and the three mental health dimensions and then keep the current H5 (i.e., the mediation effect of the three mental health dimensions in the relationship between internet use and subjective well-being). This can also help readers understand the direction and valence of each path.

There are many tutorials about how to propose direct and mediating hypotheses in research. You may consider referring to them.

Response 4-b, c: Thank you very much for the thoughtful guidance. We have carefully revised the manuscript to reduce redundancy, tighten theory–evidence–hypothesis linkages, and improve readability:

1.Embedding within the Literature Review. To avoid a stand-alone, repetitive “Hypotheses” section and to keep a clear theory→evidence→hypothesis flow, each hypothesis is now placed at the end of its corresponding Literature Review subsection: H1 after Internet Use & Subjective Well-Being; H2–H4 after Internet Use & Mental Health; H5 after Mental Health & Subjective Well-Being. We also added brief transition sentences immediately before each hypothesis to lead in the proposition smoothly.

2. Direct associations added. In line with your suggestion, we made the previously implicit direct links explicit:

•H2: Internet use is negatively associated with psychological anxiety.

•H3: Internet use is negatively associated with social loneliness.

•H4: Internet use is negatively associated with goal deficiency.

3. Mediation clarified. We retain H5 as a concise test of the overall indirect effect via the three mental-health dimensions (parallel mediation).

Manuscript revision location: Literature review, lines 176,238,281.

Comment 5-a: Measurement

a. “Variable manipulation” is often used in experiments when you manipulate a variable through stimuli. Please change it to “variable” or just remove it.

Response 5-a: Thank you for the helpful clarification. We have removed the term “variable manipulation”—which is inappropriate for an observational study—and replaced it with “variables”

Manuscript revision location: Measure, lines 403.

Comment 5-b:

b. Your conceptualization of the three dimensions of mental health and justification should be incorporated into the literature review instead of being presented in the operationalization of measurement.

Response 5-b: We are grateful for this constructive guidance. We have relocated the conceptualization and theoretical justification for the three mental-health dimensions (psychological anxiety, social loneliness, goal deficiency) from the Measures section to the Literature Review. Specifically, brief theory-anchored statements now appear in the “Internet Use and Mental Health” and “Mental Health and Subjective Well-Being” subsections to justify why these constructs are relevant mediators. The Measures section now only reports operational details (CGSS items, response scales, coding, and summary statistics), with a cross-reference to the Literature Review for conceptual grounding.

Manuscript revision location: Measure, lines 421–429.

Comment 5-c:

c.In measurement, the symbol of mean (M) and standard deviation (SD) for each variable should be italicized. Additionally, why did you not include the values of the mean and standard deviation for other variables, such as SWB, internet use, and control variables? Please go through your manuscript closely to ensure that all statistics are reported consistently.

Response 5-c: Th

---

## [Decision Letter · Decision Letter 3]

17 Sep 2025

Dear Dr Tan,

Thank you for submitting your manuscript to PLOS ONE. After careful consideration, we feel that it has merit but does not fully meet PLOS ONE’s publication criteria as it currently stands. Therefore, we invite you to submit a revised version of the manuscript that addresses the points raised during the review process.

Please revise your manuscript carefully in accordance with the reviewers’ comments. In addition, I encourage you to give close attention to the overall presentation of the paper. Specifically, please proofread thoroughly to ensure consistency in grammar, spelling, and academic style. Furthermore, double-check all references and citations for accuracy, completeness, and compliance with the journal’s formatting requirements. It is also important to ensure that the language and expressions used in the manuscript accurately reflect the nature of your research, which is based on a cross-sectional survey. In addition, please ensure that terminology is used consistently throughout the manuscript. Finally, review the formatting details—such as headings, subheadings, and spacing—to enhance clarity and readability.

We look forward to receiving your revised manuscript.

Kind regards,

Lianshan Zhang

Academic Editor

PLOS ONE

Journal Requirements:

Reviewers' comments:

Reviewer's Responses to Questions

**Comments to the Author**

Reviewer #3: All comments have been addressed

2. Is the manuscript technically sound, and do the data support the conclusions?

Reviewer #3: Yes

3. Has the statistical analysis been performed appropriately and rigorously?

Reviewer #3: Yes

4. Have the authors made all data underlying the findings in their manuscript fully available?

Reviewer #3: Yes

5. Is the manuscript presented in an intelligible fashion and written in standard English?

Reviewer #3: Yes

Reviewer #3: Thank you for your revision. The revised manuscript has been improved a lot. But I still have a few suggestions below:

1. Hypothesis 1 (H1) — "Internet use is positively associated with subjective well-being among older adults" — should be positioned prior to the paragraph beginning with “Despite these insights, existing research has primarily emphasized the outcomes of internet use on well-being, while neglecting the internal psychological processes through which such effects occur.” This paragraph emphasizes the mediating effect of mental health. Additionally, before proposing H1, please provide justification for the positive correlation between internet use and subjective well-being based on prior research.

2. In the literature review section titled “Internet Use and Mental Health”, after the sentence “The relationship between internet use and mental health among older adults is multifaceted,” please include a definition of mental health as used in your study, specifically outlining its three dimensions.

3. For Hypothesis 5 (H5) — “Internet use affects subjective well-being indirectly through mental health” — please replace “affects” with “is associated with” to avoid implying causality, as your study is based on a cross-sectional survey design.

4. Consider renaming the title of Table 1 to: “Descriptive Statistics of Key Variables” for clarity.

5. When reporting the Pearson correlation coefficients among the key study variables, you may report the non-significant results briefly and refer readers to Table 2. Since most of the correlations are significant, this approach will help reduce redundancy in the text.

6. Please move the results related to H1 — “By contrast, the direct effect of internet use on subjective well-being is not statistically significant (β = 0.059, p = 0.082), so H1 is not supported” — before the results for H2–H4 to maintain the logical flow of hypothesis testing.

7. In your discussion section, please elaborate more clearly on how the findings of this study extend, challenge, or align with existing research. This will help to better justify and highlight your theoretical contributions.

**Do you want your identity to be public for this peer review?** For information about this choice, including consent withdrawal, please see our Privacy Policy

Reviewer #3: No

---

## [Author Response · Author response to Decision Letter 4]

24 Sep 2025

Response to Academic Editor and Reviewers

Manuscript Title: The Impact of Internet Use on the Subjective Well-Being of Older Adults: The Mediating Role of Mental Health

Manuscript Number: [PONE-D-24-57555]

Dear Academic Editor and Reviewers,

We would like to express our sincere gratitude to the Academic Editor and the Reviewers for their thorough evaluation and constructive feedback on our manuscript. We greatly appreciate the time and effort they devoted to improving the clarity, conceptual grounding, and methodological rigor of this study.

In response to each comment, we have carefully revised the manuscript and prepared point-by-point replies. For clarity, each reviewer comment is restated, followed by our detailed response and the specific location of the corresponding changes in the revised manuscript.

We sincerely hope that the revised version meets your expectations and is now suitable for consideration of publication in PLOS ONE. Once again, we thank you for your valuable time and effort.

Response to Academic Editor’s Comments

Comments 1: Please revise your manuscript carefully in accordance with the reviewers’ comments. In addition, I encourage you to give close attention to the overall presentation of the paper. Specifically, please proofread thoroughly to ensure consistency in grammar, spelling, and academic style. Furthermore, double-check all references and citations for accuracy, completeness, and compliance with the journal’s formatting requirements. It is also important to ensure that the language and expressions used in the manuscript accurately reflect the nature of your research, which is based on a cross-sectional survey. In addition, please ensure that terminology is used consistently throughout the manuscript. Finally, review the formatting details—such as headings, subheadings, and spacing—to enhance clarity and readability.

Response 1: We sincerely thank the Academic Editor for these detailed comments regarding the overall presentation of the manuscript. In revising the paper, we took the following steps:

1. Proofreading and consistency:

The entire manuscript was carefully proofread to ensure consistency in grammar, spelling, and academic style. Key terminology (e.g., subjective well-being, internet use, cross-sectional survey) was reviewed and standardized throughout the text.

2. Clarification of study design:

To accurately reflect the cross-sectional nature of our research, we revised the wording in both the Abstract and Methods sections. Specifically, terms that could imply causality (e.g., “effect,” “impact”) were replaced with more precise expressions such as “association” and “relationship.”

3. References and citations:

All references were checked one by one for accuracy, completeness, and compliance with the PLOS ONE reference style, which follows the International Committee of Medical Journal Editors (ICMJE) recommendations. Journal abbreviations were verified using the NCBI database. DOI numbers were added where available, and the formatting was aligned with the official guidelines (see: http://journals.plos.org/plosone/s/submission-guidelines#loc-references).

4. Formatting details:

Headings, subheadings, and spacing were reviewed and standardized across the manuscript. Tables and figure captions were reformatted to ensure clarity, consistency, and adherence to journal standards.

5. Language refinement:

Expressions in the Abstract, Introduction, and Discussion were revised to improve clarity and academic tone, ensuring that the language accurately conveys the findings and contributions of the study.

We believe these revisions have improved the overall quality and presentation of the manuscript and respectfully submit the revised version for your consideration.

Response to Reviewer 3’s Comments:

Comments 1: Hypothesis 1 (H1) — "Internet use is positively associated with subjective well-being among older adults" — should be positioned prior to the paragraph beginning with “Despite these insights, existing research has primarily emphasized the outcomes of internet use on well-being, while neglecting the internal psychological processes through which such effects occur.” This paragraph emphasizes the mediating effect of mental health. Additionally, before proposing H1, please provide justification for the positive correlation between internet use and subjective well-being based on prior research.

Response 1: We sincerely thank the reviewer for this helpful suggestion. In response, we have repositioned Hypothesis 1 so that it now appears before the paragraph beginning with “Despite these insights…”, thereby ensuring that the logical flow proceeds from the direct association between internet use and subjective well-being to the mediating role of mental health.

In addition, we have strengthened the justification for H1 by incorporating recent empirical studies. Specifically, a nationally representative survey from China demonstrated that internet use frequency, the size of online social networks, and digital proficiency were each positively associated with subjective well-being among middle-aged and older adults [26]. Another study focusing on older adults’ online activities reported that communicative uses of the internet had stronger positive effects on eudaimonic well-being than passive or purely informational uses [27]. Comparative research conducted in Finland and Sweden further showed that socially oriented and active internet use was positively related to subjective well-being and psychological health among older populations [28].

The revised paragraph now concludes with the sentence “Collectively, these studies provide a solid basis for the following hypothesis:”, which more clearly demonstrates the empirical grounding of H1 and provides a smoother transition from the literature review to hypothesis development.

Manuscript revision location: Literature review, lines 127–138.

Comments 2: In the literature review section titled “Internet Use and Mental Health”, after the sentence “The relationship between internet use and mental health among older adults is multifaceted,” please include a definition of mental health as used in your study, specifically outlining its three dimensions.

Response 2: We are grateful to the reviewer for this insightful comment. We agree that it is essential to provide a clear definition of mental health as operationalized in our study, especially since it directly underpins the mediating variables tested in our model. Following the reviewer’s advice, we have revised the literature review section accordingly.

Specifically, immediately after the sentence “The relationship between internet use and mental health among older adults is multifaceted,” we have added the following sentence: “In this study, mental health is conceptualized as a multidimensional construct encompassing three aspects: psychological anxiety, social loneliness, and goal deficiency. These dimensions reflect emotional distress, perceived social disconnection, and a diminished sense of purpose in later life.”

This addition makes the scope of “mental health” more explicit, reduces possible ambiguity, and ensures better alignment between the conceptual framework, the empirical measures, and the hypotheses (H2–H4) presented later in the paper. We believe this change strengthens the clarity and coherence of the literature review.

Manuscript revision location: Literature review, lines 148–152.

Comments 3: For Hypothesis 5 (H5) — “Internet use affects subjective well-being indirectly through mental health” — please replace “affects” with “is associated with” to avoid implying causality, as your study is based on a cross-sectional survey design.

Response 3: We thank the reviewer for this important clarification. We fully agree that causal expressions should be avoided given the cross-sectional design of our study. Following the reviewer’s suggestion, we have carefully revised Hypothesis 5 and all related descriptions throughout the manuscript.

Specifically, the original statement “H5: Internet use affects subjective well-being indirectly through mental health” has been modified to “H5: Internet use is associated with subjective well-being indirectly through mental health.” Corresponding expressions in the results and discussion sections have also been adjusted. For instance:

• In the mediation results, “indicating that internet use influences well-being primarily through mental-health pathways” has been revised to “indicating that internet use is associated with well-being primarily through mental-health pathways.”

• In the concluding synthesis, “mental-health variables collectively mediate the association between internet use and SWB” has been retained, but causal wording has been avoided by rephrasing the internal pathway as “digital engagement is associated with well-being largely through its links with lower anxiety and stronger goal-related motivation.”

Manuscript revision location: Literature review, line 231.

Comments 4: Consider renaming the title of Table 1 to: “Descriptive Statistics of Key Variables” for clarity.

Response 4: We appreciate the reviewer’s helpful suggestion. We have revised the title of Table 1 from “Basic variable description statistics table” to “Descriptive Statistics of Key Variables” to improve clarity and better align with academic writing conventions.

Manuscript revision location: Results, line 325.

Comments 5: When reporting the Pearson correlation coefficients among the key study variables, you may report the non-significant results briefly and refer readers to Table 2. Since most of the correlations are significant, this approach will help reduce redundancy in the text.

Response 5: We thank the reviewer for this constructive suggestion. We agree that presenting all correlation coefficients in detail within the text created unnecessary redundancy with Table 2. Following the reviewer’s advice, we have revised the correlation analysis section to provide a more concise description.

In the revised version, we highlight only the key significant associations by reporting representative coefficients (e.g., the positive correlation between internet use and subjective well-being, r = .10, p < .01, and the strongest negative correlation between psychological anxiety and subjective well-being, r = –.27, p < .01). Other significant correlations are summarized narratively, while non-significant correlations are briefly noted and referred to Table 2. This adjustment reduces redundancy, improves readability, and ensures that Table 2 remains the main reference for detailed coefficients.

Manuscript revision location: Results, lines 328–338.

Comments 6: Please move the results related to H1 — “By contrast, the direct effect of internet use on subjective well-being is not statistically significant (β = 0.059, p = 0.082), so H1 is not supported” — before the results for H2–H4 to maintain the logical flow of hypothesis testing.

Response 6: We sincerely thank the reviewer for pointing out this issue with the presentation order. We agree that the results for H1 should logically precede those for H2–H4 to ensure consistency with the sequence of hypothesis testing. Accordingly, we have revised the Results section so that the finding regarding H1—the non-significant direct effect of internet use on subjective well-being (β = 0.059, p = 0.082)—is now presented before the results for H2–H4. This adjustment improves the logical flow of the results while leaving the statistical findings unchanged.

Manuscript revision location: Results, lines 420–429.

Comments 7: In your discussion section, please elaborate more clearly on how the findings of this study extend, challenge, or align with existing research. This will help to better justify and highlight your theoretical contributions.

Response 7: We sincerely thank the reviewer for this constructive suggestion. In response, we have carefully revised the Discussion section (lines 81–240) to articulate more explicitly how our findings extend, align with, and challenge existing research, thereby highlighting the theoretical contributions of the study.

1. Extending prior work: By incorporating goal deficiency as an additional mediator alongside psychological anxiety and social loneliness, our study broadens the explanatory framework of how digital engagement influences well-being in later life. This extends previous studies that predominantly examined anxiety or loneliness in isolation.

2. Aligning with existing evidence: We explicitly situate our results within the “indirect-only” mediation model, emphasizing that the benefits of internet use for older adults’ well-being arise mainly through psychological or social processes rather than direct associations [50,51].

3. Challenging common assumptions: We note that H1 was not supported, as the direct link between internet use and subjective well-being was non-significant. This challenges the conventional expectation of a straightforward positive relationship and underscores the necessity of considering intermediary psychological mechanisms.

Furthermore, we integrate our findings with recent stratification research, which demonstrates that the impact of internet use on well-being is heterogeneous across subjective social class and is largely transmitted through psychological pathways [54]. Collectively, these revisions allow the Discussion to more clearly demonstrate that digital participation enhances older adults’ well-being primarily by reinforcing psychological resources and sense of purpose, rather than exerting direct effects.

We are grateful for this suggestion, which has helped us refine the discussion and present our theoretical contributions in a clearer and more coherent manner.

Manuscript revision location: Discussion, lines 451–516.

Once again, we would like to express our sincere gratitude to the academic editor and reviewers for your constructive suggestions and valuable guidance. Your insights have been instrumental in improving the quality, clarity, and rigor of this manuscript. We hope that the revised version has adequately addressed all the comments and is now suitable for consideration for publication in PLOS ONE. Should there be any further questions or revisions required, we will be more than willing to cooperate. Thank you again for your time and kind consideration.

With kind regards,

Chunyun Tan

Corresponding Author (on behalf of all authors)

---

## [Editor Report · Decision Letter 4]

20 Oct 2025

The Impact of Internet Use on the Subjective Well-Being of Older Adults: The Mediating Role of Mental Health

PONE-D-24-57555R4

Dear Chun Yun Tan,

We’re pleased to inform you that your manuscript has been judged scientifically suitable for publication and will be formally accepted for publication once it meets all outstanding technical requirements.

Kind regards,

Lianshan Zhang

Academic Editor

PLOS ONE
---

## [Editor Report · Acceptance letter]

PONE-D-24-57555R4

PLOS ONE

Dear Dr. Tan,

I'm pleased to inform you that your manuscript has been deemed suitable for publication in PLOS ONE. Congratulations! Your manuscript is now being handed over to our production team.

Kind regards,

on behalf of

Dr. Lianshan Zhang

Academic Editor

PLOS ONE